# Textual Stochastic Gradient Descent: Discrete Optimization of External Memory for Reasoning Language Agents

**Jian Li** [1 2 3]   **Hua Huang** [1 2 3]

## Abstract

While Large Language Models (LLMs) possess strong reasoning capabilities, enabling them to learn continuously from experience without parametric retraining remains an open challenge. Existing Retrieval-Augmented Generation (RAG) approaches typically treat memory as a static or append-only corpus, leading to "memory saturation", where accumulating noise and redundant information degrade performance over time. To address this, we propose an Experience Library Optimization framework that treats the agent's external memory, which we call the experience library, as a learnable parameter under an explicit capacity budget. We introduce Textual Stochastic Gradient Descent (TSGD), a discrete optimization algorithm that refines this library via failure-driven Add, Edit, and Delete operations. TSGD estimates "textual gradients" through self-reflection and uses a dual-verification mechanism to ensure generalization, which prevents overfitting to local errors. Empirical results on MATH and AIME benchmarks show that TSGD achieves state-of-the-art performance, improving accuracy by up to 18.7% over zero-shot baselines and substantially outperforming static RAG, while keeping a compact memory footprint (compressing hundreds of experiences into $\approx$ 30 high-utility rules).

## 1. Introduction

Learning from experience and adapting to new problems is a basic feature of intelligence. For Large Language Model

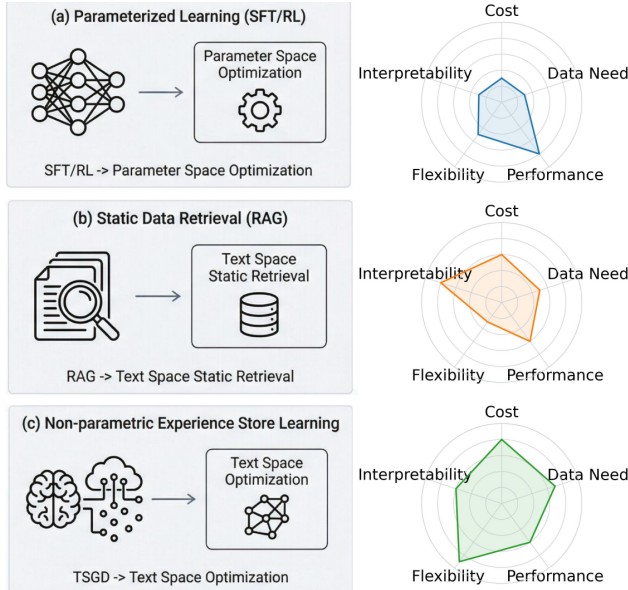

*Figure 1.* The evolution of agent learning paradigms. We transition from implicit parameter updates (SFT/RL) and static retrieval (RAG) to *Experience Library Learning*, where the memory itself is actively optimized in text space under a strict capacity budget.

(LLM) agents, this means not only reasoning within a fixed context but also accumulating and refining knowledge over time (Park et al., 2023; Wang et al., 2024a; Packer et al., 2023). Two paradigms currently dominate, and both have clear limitations.

The first, parametric learning, updates the model's weights through Supervised Fine-Tuning (SFT) (Ouyang et al., 2022) or Reinforcement Learning (RL) (Schulman et al., 2017; Shao et al., 2024). It works well for internalizing general skills, but it is computationally expensive, slow to iterate, and prone to catastrophic forgetting, where new knowledge displaces what the model already knew in dynamic, multi-domain settings.

The second, non-parametric memory such as RAG (Lewis et al., 2020; Guu et al., 2020; Sarthi et al., 2024; Asai et al., 2024), moves knowledge into an external retrieval corpus. This avoids forgetting and supports fast updates. Most existing systems, however, treat the memory as a static database

[1]School of Artificial Intelligence, Beijing Normal University, Beijing, China [2]Beijing Key Laboratory of Artificial Intelligence for Education, Beijing, China [3]Engineering Research Center of Intelligent Technology and Educational Application, Ministry of Education, Beijing, China. Correspondence to: Hua Huang <huahuang@bnu.edu.cn>.

*Proceedings of the 43$^{rd}$ International Conference on Machine Learning*, Seoul, South Korea. PMLR 306, 2026. Copyright 2026 by the author(s).

or an append-only cache with no quality control. Because they have no optimization mechanism, false positives accumulate, obsolete entries persist, and redundant items crowd out the context window. As the library grows its signal-to-noise ratio drops, producing what we call "memory saturation", where more experience can actually hurt performance.

We argue that the missing piece is the optimization of non-parametric memory. Just as we reduce loss by updating model parameters, we can reduce task error by updating the contents of the memory itself. Figure 1 summarizes this progression, from parameter optimization (SFT/RL) to static retrieval (RAG) and then to experience library learning. Here the learnable object is the agent's own external memory, a compact natural-language store of reusable strategies that we call the experience library. We therefore ask:

> *Under a frozen base model $\mathcal{M}_\theta$, can we treat a natural-language experience library $\Phi$ as a learnable object, optimizing it to maximize expected task utility under a strict capacity budget?*

Optimizing an experience library raises two difficulties. The first is that the search space is discrete and non-differentiable: unlike continuous weights, text experiences cannot be updated by backpropagation, so the "gradient", the direction of improvement, has to be inferred semantically (Yuksekgonul et al., 2025). The second is the tension between local repair and global capacity. Appending a correction for every error eventually causes memory saturation, so a good optimizer has to weigh immediate error correction against the long-term health of the library, keeping entries that generalize rather than overfit.

We address these difficulties by formalizing the task as Experience Library Optimization under an explicit capacity budget. We propose Textual Stochastic Gradient Descent (TSGD)[1], a framework that iteratively refines $\Phi$ through discrete Add, Edit, and Delete operations. Each update follows a "textual gradient" that an LLM estimates by reflecting on a failure and identifying its root cause. A dual verification step then accepts an update only when it fixes the current error without hurting retrieval quality elsewhere, and a periodic regularization step compresses the library by merging redundant entries so that its signal-to-noise ratio stays high. On MATH and AIME, TSGD clearly outperforms static RAG and prompt-based baselines, generalizing better while using a much smaller memory.

This paper makes three contributions. We cast experience library learning as Experience Library Optimization, treating a natural-language memory $\Phi$ as the object to optimize under an explicit capacity budget. We then introduce TSGD,

a discrete text-space optimizer that performs failure-driven Add, Edit, and Delete updates with dual verification and periodic regularization. Finally, we report consistent gains on MATH and AIME, particularly under distribution shift, while keeping the library compact; the appendix adds cross-domain results on code generation (HumanEval/MBPP) and tool use (ToolBench), as well as open-source backbones (Qwen-2.5-72B and Llama-3.1-70B).

## 1.1. Related Work

We organize related work into four parts: retrieval-augmented generation and external memory, example/trajectory-based reasoning augmentation, self-reflection and correction, and the view of memory as an optimizable non-parametric object.

**Retrieval-augmented generation and external memory.** RAG and external memory improve factuality and reasoning by retrieving relevant text or vectors as context for a frozen (or weakly updated) base model (Lewis et al., 2020; Borgeaud et al., 2022). Recent advances focus on structural understanding via graph-based retrieval (Edge et al., 2024) or robustness against irrelevant contexts through corrective mechanisms (Yan et al., 2024). However, many systems treat the memory as static data or a write-when-needed cache whose quality is controlled by heuristics; as the library grows, noise and topic drift degrade retrieval, matching our empirical observation of capacity saturation.

**Example and trajectory based reasoning.** In-context learning and chain-of-thought prompting treat curated demonstrations or reasoning traces as reusable experience (Wei et al., 2022; Wang et al., 2023). While scaling to many-shot regimes shows promise (Agarwal et al., 2024), finding the optimal set of examples remains challenging. In contrast, we treat experiences as editable atomic units and iteratively refine them under failure-driven updates, pushing them from explaining one instance to transferable strategies or theorem application patterns.

**Reflection, correction, and verification.** Recent methods reduce reasoning errors by generating critiques or using external tools for verification (Madaan et al., 2023; Shinn et al., 2023; Gou et al., 2024). Textual-feedback optimizers such as TextGrad (Yuksekgonul et al., 2025) mainly optimize prompts or instructions, whereas TSGD optimizes a structured experience library whose entries are retrieved, edited, and regularized under a capacity budget. However, the efficacy of intrinsic self-correction without external signals is debated, with studies suggesting performance may degrade or hallucinate fixes (Huang et al., 2024). Our dual verification explicitly decides whether an edit is written into $\Phi$ by combining local repair and global constraints, avoiding

---

[1]Code is available at https://github.com/superlj666/TSGD.

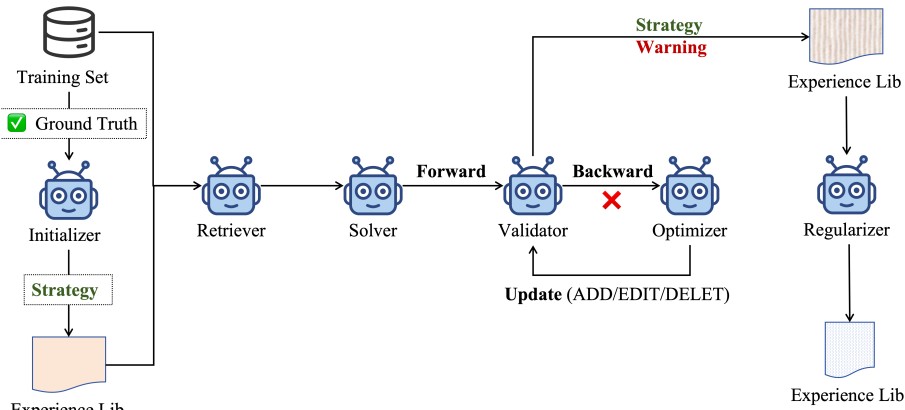

*Figure 2.* Architecture overview of Textual Stochastic Gradient Descent (TSGD). The experience library $\Phi$ is a shared state across five agents: retrieval, solver, initializer, optimizer, and regularizer.

the pitfalls of unverified self-correction.

**Optimizing non-parametric memory.** Work on model editing, compressing memory stores, and risk minimization views of external storage suggests that deciding what to write, keep, and delete is a constrained selection problem (De Cao et al., 2021; Meng et al., 2023). A closely related line of recent work also treats agent memory as an evolving object: AgentKB (Tang et al., 2025) maintains a knowledge base via heuristic add/edit/evict actions; ReasoningBank (Ouyang et al., 2026) appends and post-hoc filters experiences from self-evolving agents; and the contemporaneous FLEX (Cai et al., 2025b) performs forward learning from experience without formal capacity guarantees. Other directions include long-horizon reasoning agents (Gao et al., 2025), training-free group optimization (Cai et al., 2025a), and workflow memory (Wang et al., 2024b; Xu et al., 2025; Xiong et al., 2025). The key conceptual differences are that Experience Library Optimization formalizes the library as the optimization variable under a strict capacity budget $B$, and TSGD performs failure-driven editing with regression-tested acceptance; a detailed point-by-point comparison is provided in Appendix E.

## 2. Problem Formulation: Experience Library Optimization

We view the experience library $\Phi$ as a *controllable external conditional variable* acting upon a fixed base model $\mathcal{M}_\theta$. This perspective enables the establishment of a unified Experience Library Optimization framework.

Consider a task distribution $(q, y) \sim \mathcal{D}$, where $q$ is the query (e.g., a math problem) and $y$ is the target output. We employ a fixed-parameter agent $\mathcal{M}_\theta$ (where $\theta$ denotes the frozen weights) and a retriever $\pi_{\text{retr}}$. Given an experience library $\Phi$, for each query $q$, the agent predicts based on the retrieved

experience subset $\Phi_q = \pi_{\text{retr}}(q, \Phi)$:

$$\hat{y} = \mathcal{M}_\theta(q, \Phi_q). \tag{1}$$

The goal of experience library learning is to minimize the expected risk:

$$\min_{\Phi \in \mathcal{F}_B} R(\Phi) = \mathbb{E}_{(q,y) \sim \mathcal{D}}\big[\ell\big(\mathcal{M}_\theta(q, \pi_{\text{retr}}(q, \Phi)), y\big)\big] \\ + \lambda\,\Omega(\Phi), \tag{2}$$

where $\mathcal{F}_B$ represents the hypothesis space of experience libraries subject to a capacity budget constraint (i.e., $|\Phi| \leq B$). The term $\ell(\cdot, \cdot)$ denotes a task-specific loss function (e.g., 0-1 loss for correctness or negative log-likelihood), while $\Omega(\Phi)$ serves as a regularization term penalizing library complexity or redundancy, weighted by $\lambda$.

**Relation to prior memory-augmented agents.** Experience Library Optimization offers a unified view of recent advances. Common practices such as manual prompt engineering (Wei et al., 2022), Retrieval-Augmented Generation (RAG) (Lewis et al., 2020; Guu et al., 2020), reflection-based memory (Shinn et al., 2023; Madaan et al., 2023), and memory-augmented agents (e.g., MemGPT; Packer et al., 2023) can all be viewed as approximate solutions to the same objective $R(\Phi)$ under different parameterizations and optimizer choices. While traditional methods often treat $\Phi$ as static or manually curated, our approach directly optimizes $\Phi$ as a learnable object.

### 2.1. Formal Definitions

To make the rest of the paper self-contained, we collect the formal definitions used by TSGD.

**Definition 2.1** (Experience library). An *experience library* is a finite set $\Phi = \{e_1, \ldots, e_n\}$ where each $e_i \in \mathcal{V}^*$ is a natural-language string in some vocabulary $\mathcal{V}$ representing

a reusable, condition–strategy pair. The *capacity-bounded* hypothesis class is $\mathcal{F}_B = \{\Phi : |\Phi| \leq B\}$.

**Definition 2.2** (Textual gradient). Given a failed instance $(q, y, \hat{y})$ and retrieved subset $\Phi_q$, the *textual gradient* is a triplet $a = (\text{OP}, \text{target}, \text{content})$ produced by an optimizer agent $\pi_{\text{opt}}$, where $\text{OP} \in \{\text{ADD}, \text{EDIT}, \text{DELETE}\}$, $\text{target} \in \Phi \cup \{\varnothing\}$, and $\text{content} \in \mathcal{V}^*$. The induced update is $\Phi' = \text{Apply}(a, \Phi)$.

**Definition 2.3** (Dual verification). Given a candidate update $a$ with current and candidate libraries $\Phi, \Phi'$, a tolerance $\epsilon \geq 0$, and a validation set $S_{\text{val}}$, *local verification* returns $V_{\text{local}}(a, q_i) = \mathbb{I}[\mathcal{M}_\theta(q_i, \pi_{\text{retr}}(q_i, \Phi')) = y_i]$ and *global verification* returns $V_{\text{global}} = \text{Score}(\Phi', S_{sub}) - \text{Score}(\Phi, S_{sub})$ on a content-aware subset $S_{sub} \subseteq S_{\text{val}}$. The update is accepted iff $V_{\text{local}} = 1$ and $V_{\text{global}} > -\epsilon$.

**Definition 2.4** (Regularization budget). A *regularization budget* $B$ is an upper bound on $|\Phi|$. The regularizer agent $\pi_{\text{reg}}$ enforces $|\Phi| \leq B$ via hierarchical clustering, semantic merging, and global pruning.

Directly solving Eq. (2) is challenging due to (i) the discrete and non-differentiable nature of $\Phi$, which precludes standard gradient-based optimization, and (ii) the combinatorial search space involved in finding the optimal set of experiences. To address these, we propose TSGD, which approximates gradient descent in the semantic space.

# 3. The TSGD Framework

We present *Textual Stochastic Gradient Descent* (TSGD), a framework that operationalizes experience learning through a collaborative multi-agent system. At the core of TSGD is the dynamic evolution of a discrete experience library $\Phi$, functioning as a learnable, non-parametric memory. In the following, we first delineate the functional roles of the constituent agents and then detail the TSGD algorithm.

## 3.1. Agent Roles and Collaborative Lifecycle

Our framework employs six agents that collaborate to construct, retrieve, and refine the experience library $\Phi$.

**Initializer Agent ($\pi_{\text{init}}$)** The initializer distills atomic experiences from ground-truth (GT) trajectories. Given a query-answer pair $(q, y)$, it extracts an experience $e_{\text{init}} = \{\text{Condition} \rightarrow \text{Strategy}\}$. To ensure generalizability, $\pi_{\text{init}}$ is tasked with *de-particularization*—stripping problem-specific entities to transform concrete solutions into abstract, high-level heuristics.

**Retrieval Agent ($\pi_{\text{retr}}$)** The retrieval agent identifies the $k$ most relevant strategies from $\Phi$ for a query $q$. It implements a two-stage hybrid process: 1) *Hard Filtering*: narrowing the search space based on metadata (e.g., subject,

difficulty) to ensure domain alignment when such metadata is available; 2) *Dense Recall*: computing semantic similarity $\text{sim}(\phi(q), \phi(e))$ using an embedding model $\phi(\cdot)$ to select the top-$N$ candidates, yielding a task-specific subset $\Phi_q \subset \Phi$. In open-domain settings without structured metadata, hard filtering is bypassed and the agent relies solely on dense recall (see Appendix F).

**Solver Agent ($\pi_{\text{solver}}$)** The solver orchestrates task execution by conditioning on the retrieved experiences. Taking $q$ and $\Phi_q$ as input, it generates a reasoning trajectory $\tau$ and the final prediction $\hat{y}$.

**Validator Agent ($\pi_{\text{validator}}$)** The validator plays two roles. At *inference* time, it serves as a validity check on the solver's output (verifying that the trajectory $\tau$ and prediction $\hat{y}$ are well-formed and satisfy problem constraints). At *optimization* time, it executes the *Dual Verification* protocol (Definition 2.3) that decides whether a proposed update is written into $\Phi$.

**Optimizer Agent ($\pi_{\text{opt}}$)** The optimizer analyzes reasoning failures to derive a *textual gradient*—a semantic recommendation for library modification. Given $(q, y, \hat{y}, \Phi_q)$, it diagnoses the failure as either "missing knowledge" (Add) or "erroneous experience" (Edit/Delete), outputting a discrete operation $a \in \{\text{ADD}, \text{EDIT}, \text{DELETE}\}$ together with its associated textual content.

**Regularizer Agent ($\pi_{\text{reg}}$)** The regularizer maintains the library's signal-to-noise ratio and ensures compliance with the capacity constraint $B$. It performs hierarchical pruning:

- **Subject-Level Clustering**: partitioning $\Phi$ into domain-specific clusters.

- **Semantic Merging**: consolidating redundant experiences within each cluster.

- **Global Pruning**: removing low-utility experiences across clusters to optimize library size.

## 3.2. Textual Stochastic Gradient Descent (TSGD)

We propose TSGD (Algorithm 1) to optimize $\Phi$ by minimizing the empirical risk $\hat{R}_\mathcal{D}(\Phi)$. TSGD bridges the gap between discrete prompt optimization and traditional gradient descent by utilizing LLM-generated feedback as a surrogate for the gradient. Unlike standard SGD which operates in a continuous parameter space, TSGD performs updates in a discrete textual space, where the "gradient" is a semantic instruction for library refinement.

**Parallel Asynchronous Optimization** To facilitate large-scale training, TSGD processes samples $q_i \in S_{\text{train}}$ via an

**Algorithm 1** Textual Stochastic Gradient Descent (TSGD)

**Input:** Training data $S_{\text{train}}$, Validation data $S_{\text{val}}$, Budget $B$, Tolerance $\epsilon$
**Output:** Optimized experience library $\Phi^*$
**Initialize:** $\Phi \leftarrow \pi_{\text{init}}(S_{\text{seed}})$
**for** each $q_i \in S_{\text{train}}$ **in parallel do**
    {Step 1: Retrieval}
    $\Phi_{q_i} \leftarrow \pi_{\text{retr}}(q_i, \Phi)$
    {Step 2: Forward Pass}
    $\hat{y}_i, \tau_i \leftarrow \pi_{\text{solver}}(q_i, \Phi_{q_i})$
    **if** $\hat{y}_i \neq y_i$ **then**
        {Step 3: Backward Pass}
        $a \leftarrow \pi_{\text{opt}}(q_i, y_i, \hat{y}_i, \Phi_{q_i})$ {Textual gradient estimation}
        {Step 4: Update with Dual Verification}
        $\Phi' \leftarrow \text{Apply}(a, \Phi)$
        $V_{\text{local}}, V_{\text{global}} \leftarrow \pi_{\text{validator}}(a, \Phi, \Phi', q_i, S_{\text{val}})$
        **if** $V_{\text{local}}$ **is** True **and** $V_{\text{global}} > -\epsilon$ **then**
            $\Phi \leftarrow \Phi'$ {Accept the update}
        **end if**
    **end if**
    **if** periodic **then** $\Phi \leftarrow \pi_{\text{reg}}(\Phi)$ {Regularization}
**end for**
**return** $\Phi$

asynchronous parallel workflow. While Algorithm 1 depicts a sequential view, the implementation employs a concurrency control mechanism to maintain library integrity. Specifically, threads acquire a *Read Lock* during retrieval and forward passes to ensure a consistent view of $\Phi$, and a *Write Lock* to perform atomic modifications when an update is accepted. This architecture enables $\Phi$ to evolve dynamically as it encounters diverse failure modes across the training distribution.

**Dual Verification Mechanism** A primary challenge in discrete optimization is preventing overfitting to individual samples. TSGD addresses this through the Validator Agent ($\pi_{\text{validator}}$), which enforces a two-stage verification process (Definition 2.3):

- **Local Verification**: the agent confirms that the proposed operation $a$ successfully rectifies the failure on the current query $q_i$. This ensures the "textual gradient" is locally valid.

- **Global Verification**: to ensure generalization, the agent constructs a *content-aware validation set* $S_{sub} \subset S_{\text{val}}$ by retrieving validation samples semantically similar to $q_i$ (typically $|S_{sub}| \approx 5\text{–}10$, avoiding the full-validation-set overhead). It calculates the performance gain $\Delta = \text{Score}(\Phi', S_{sub}) - \text{Score}(\Phi, S_{sub})$. An update is accepted only if $\Delta$ exceeds a tolerance threshold

$-\epsilon$, allowing for minor regressions if they are compensated by broader improvements.

The cost of global verification is therefore $O(|S_{sub}|)$ per accepted candidate (a few extra LLM calls), not $O(|S_{\text{val}}|)$; a detailed cost decomposition is given in Appendix H.

**Periodic Regularization** As $\Phi$ evolves, redundancy and noise may accumulate, potentially leading to "memory saturation" and reduced retrieval efficiency. The regularizer agent $\pi_{\text{reg}}$ is invoked periodically (e.g., after every $N = 50$ samples) to maintain the library's signal-to-noise ratio. By performing hierarchical clustering and merging (as detailed in Section 3.1), $\pi_{\text{reg}}$ prunes $\Phi$ to respect the capacity constraint $B$, ensuring that only the most general and high-utility experiences are retained.

## 4. Theoretical Analysis

We provide a theoretical characterization of TSGD, focusing on its generalization properties, convergence stability, and an idealized sample-efficiency model. We also provide a representation-learning interpretation via the Information Bottleneck principle. Detailed proofs for all theorems are provided in the Appendix.

### 4.1. Generalization Bound via Capacity Control

We first analyze how the capacity constraint $B$ on the experience library $\Phi$ governs the generalization gap. We define the hypothesis class induced by capacity-bounded libraries as $\mathcal{H}_B = \{h_\Phi : \Phi \subset \mathcal{V}^*, |\Phi| \leq B\}$, where each $h_\Phi$ represents the mapping from a query $q$ to a prediction $\hat{y}$ given the retrieved subset $\Phi_q$.

**Theorem 4.1** (Generalization Bound). *For any distribution $\mathcal{D}$, with probability at least $1 - \delta$ over a training set of size $m$, the population risk $R(\Phi)$ satisfies*

$$R(\Phi) \leq \hat{R}_S(\Phi) + \mathcal{O}\left(\sqrt{\frac{B \log |\mathcal{V}^*|}{m}}\right) + \sqrt{\frac{\log(1/\delta)}{2m}}, \quad (3)$$

*where $\hat{R}_S(\Phi)$ is the empirical risk on the training set $S$.*

Theorem 4.1 provides a formal justification for the *capacity constraint $B$* implemented by the regularizer agent $\pi_{\text{reg}}$. The bound shows that the generalization error scales with $\sqrt{B/m}$. This implies that while a larger library might reduce the empirical risk $\hat{R}_S(\Phi)$ by covering more edge cases, it also increases the risk of overfitting (over-optimization) to the training instances. The regularizer agent's role in maintaining $\Phi$ within the budget $B$ is essential for bounding the generalization gap, particularly in low-data regimes.

*Table 1.* Performance comparison on MATH sub-datasets (in-distribution; Mean@32) and AIME 2024/2025 (out-of-distribution). TSGD consistently outperforms baselines in both settings. Within each group of seven columns under MATH, the best per-row sub-dataset score is bolded for TSGD rows.

| Base Model | Method | MATH Level 5 (in-distribution) | | | | | | | | OOD | |
| | | Int. Alg. | Algebra | Prealg. | Geometry | Num. Theory | Precalc. | Prob. | Overall | AIME 24 | AIME 25 |
|---|---|---|---|---|---|---|---|---|---|---|---|
| Grok-4.1 | LLM | 16.73 | 75.26 | 84.51 | 15.43 | 83.40 | 33.20 | 52.24 | 49.30 | 41.46 | 31.77 |
| | ReAct | 25.0 | 70.0 | 78.0 | 20.0 | 75.0 | 38.0 | 55.0 | 48.0 | 38.0 | 30.0 |
| | RAG | 28.0 | 72.0 | 80.0 | 22.0 | 78.0 | 40.0 | 57.0 | 50.0 | 39.0 | 31.0 |
| | TF-GRPO | 22.0 | 78.0 | 86.0 | 25.0 | 85.0 | 42.0 | 58.0 | 53.0 | 43.0 | 34.0 |
| | **TSGD** | **35.0** | **88.0** | **92.0** | **31.25** | **90.0** | **56.42** | **65.0** | **68.0** | **55.0** | **45.0** |
| GPT-4o mini | LLM | 27.65 | 85.78 | 74.82 | 12.24 | 99.72 | 44.53 | 42.58 | 54.60 | 15.21 | 12.43 |
| | **TSGD** | **45.0** | **95.0** | **85.0** | **21.65** | **99.9** | **62.31** | **55.0** | **68.0** | **30.0** | **25.0** |

## 4.2. Convergence of Textual Stochastic Gradient Descent

Optimizing in a discrete textual space presents challenges for stability. We analyze the convergence of the TSGD update sequence $\{\Phi_t\}$ under the dual verification mechanism.

**Theorem 4.2** (Monotonic Improvement). *Assume the utility function $U(\Phi) = 1 - R(\Phi)$ is bounded. If the dual verification protocol ensures that any accepted update $a_t$ satisfies $\mathbb{E}[U(\Phi_{t+1}) - U(\Phi_t) \mid accept] \geq \gamma$ for some $\gamma > 0$, then the expected utility $\mathbb{E}[U(\Phi_t)]$ converges.*

This result highlights the necessity of the *Validator Agent* ($\pi_{\text{validator}}$). In discrete optimization, a "textual gradient" $a$ derived from a single failure might be noisy or locally biased. The dual verification (local and global) acts as a *stochastic filter* that only admits updates with a high probability of providing a positive expected gain. By enforcing this monotonic property, TSGD avoids the oscillatory behavior often seen in prompt engineering and ensures that the library evolves toward a stable, high-utility state.

## 4.3. An Idealized Sample-Complexity Model for Compositional Reasoning

One of the conceptual advantages of experience learning over full-parameter fine-tuning is sample efficiency. We formalize a simple, idealized model of this phenomenon. We state this as a proposition (rather than a theorem) under an explicit assumption that abstracts away the noise of real training experiences.

**Assumption 4.3** (Idealized skill sampling). Suppose tasks are compositional over $K$ atomic skills $\{s_1, \ldots, s_K\}$ and there exists a reusable atomic experience $e_i$ for each skill $s_i$ such that the solver can compose any subset of $\{e_1, \ldots, e_K\}$ in-context to solve unseen combinations. We further assume that (a) each training experience cleanly reveals one missing skill (when the optimizer fires), and (b) repeated exposure to the same revealed skill suffices for it to be acquired by the library.

**Proposition 4.4** (Sample efficiency under Assumption 4.3).

*Under Assumption 4.3, while learning a generic $f : \mathcal{Q} \to \mathcal{Y}$ without exploiting the compositional structure can require sample complexity $\Omega(2^K)$ to distinguish among combinations, the expected sample complexity for TSGD to cover all $K$ atomic skills via the optimizer is $\mathcal{O}(K \log K)$, by analogy to the coupon collector's problem.*

Proposition 4.4 explains *why* TSGD can achieve strong results with limited training data in benign regimes: once $\pi_{\text{init}}$ or $\pi_{\text{opt}}$ identifies an atomic heuristic (e.g., a specific geometric theorem), the solver can apply it to any future combination. In practice, training experiences are partial and noisy (multiple skills may be implicated by a single failure), so Assumption 4.3 should be interpreted as an idealized model rather than a tight account of practical sample efficiency. Theorems 4.1 and 4.2 remain our primary load-bearing results.

## 4.4. Information Bottleneck Interpretation

We can further interpret the collaborative roles of the optimizer ($\pi_{\text{opt}}$) and regularizer ($\pi_{\text{reg}}$) through the Information Bottleneck (IB) principle. The optimization of $\Phi$ can be viewed as

$$\min_{\Phi} \mathcal{L}_{\text{IB}} = -I(Y; \Phi) + \beta I(Q; \Phi), \quad (4)$$

where $I(\cdot; \cdot)$ denotes mutual information.

This perspective clarifies the "textual gradient" dynamics. Maximizing $I(Y; \Phi)$ corresponds to the *Optimizer Agent*'s objective: adding predictive content to $\Phi$ to resolve reasoning failures. Conversely, minimizing $I(Q; \Phi)$ corresponds to the *Regularizer Agent*'s objective: abstracting away instance-specific details (nuisances) from the queries to prevent the library from becoming a collection of "lookup tables". The budget $B$ acts as a hard bottleneck that forces $\pi_{\text{reg}}$ to retain only the most informative, generalizable strategies, approximating a minimal sufficient statistic for the reasoning task.

*Table 2.* Ablation results on MATH Precalculus (Level 5). Full TSGD achieves superior performance while maintaining the most compact experience library. All four pairwise comparisons against the LLM baseline have $p < 10^{-4}$ under McNemar's test ($n$=1018); see Appendix J.

| Method | Pass@1 | Pass@2 | Pass@4 | Pass@8 | Exp. Cap. ($B$) |
|---|---|---|---|---|---|
| LLM (Baseline) | 33.52 | 49.06 | 63.73 | 76.87 | – |
| + Initialization | 55.67 | 70.77 | 84.09 | 93.26 | 533 |
| + Optimization | 47.21 | 65.45 | 81.35 | 93.65 | 662 |
| + Verification | 53.61 | 73.40 | 90.06 | 98.08 | 204 |
| **TSGD (Full)** | **58.33** | **76.88** | **90.68** | **97.22** | **27** |

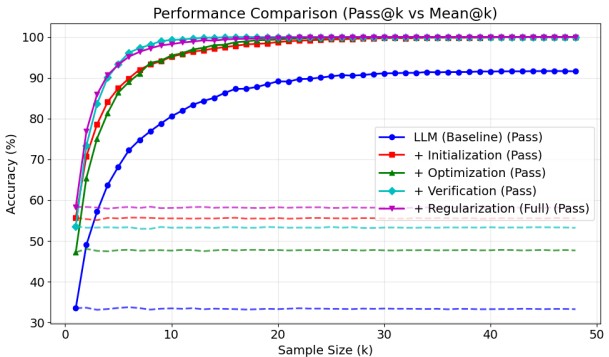

*Figure 3.* Ablation performance trends across configurations.

# 5. Experiments

We evaluate TSGD along two dimensions: (i) in-distribution performance on high-difficulty MATH subsets, and (ii) OOD generalization on AIME 2024 and 2025. In the appendix we additionally report cross-domain results on HumanEval, MBPP, and ToolBench (Appendix F), evaluation on open-source backbones Qwen-2.5-72B and Llama-3.1-70B (Appendix G), conceptual comparison with AgentKB and related editable-memory systems (Appendix E), empirical comparisons with AWM, ACE, and FLEX (Appendix H), and significance tests for the ablation comparisons (Appendix J).

## 5.1. Performance Comparison

### 5.1.1. EXPERIMENTAL SETUP

We evaluate TSGD on high-difficulty reasoning tasks: (i) in-distribution performance on MATH Level 5 subsets, and (ii) Out-of-Distribution (OOD) generalization on AIME 2024 and 2025. We use 2,097 MATH training samples (60% of the original split) to construct the experience library. Baseline comparisons in the main table include zero-shot LLM, ReAct-style reasoning, RAG (retrieving ground-truth trajectories), and TF-GRPO (training-free GRPO) where available. We report Mean@32 accuracy as the primary metric. Empirical comparisons against AWM, ACE, and FLEX under matched conditions (same backbone, problem

set, prompt budget, and evaluation harness) are reported in Appendix H; AgentKB is discussed conceptually in Appendix E.

### 5.1.2. MAIN RESULTS

Table 1 summarizes the performance across different base models and subjects. On Grok-4.1 Fast Non-Reasoning, TSGD achieves 68.0% overall accuracy on MATH Level 5, a +18.7% improvement over the zero-shot baseline and a gain over the reported retrieval and prompt-based baselines. The gains are particularly significant in Geometry (+15.82%) and Intermediate Algebra (+18.27%), where domain-specific heuristics are crucial. Notably, TSGD exhibits strong OOD generalization on AIME, suggesting that the learned textual gradients capture fundamental reasoning patterns rather than memorizing training instances. Results on GPT-4o mini further confirm the model-agnostic effectiveness of TSGD relative to the zero-shot setting. Results on open-source backbones (Qwen-2.5-72B and Llama-3.1-70B) are reported in Appendix G.

**Cost considerations.** TSGD's per-query *retrieval* latency is identical to standard RAG (a CPU-side embedding search). The *prompt-token* budget per query is also comparable to RAG in our setup, since TSGD retrieves a small set of distilled rules ($k \leq 16$) rather than full raw trajectories. However, TSGD incurs a one-time offline construction cost: approximately 120–124 optimization calls excluding seeding, or about 162 total calls including seeding and periodic regularization in the 20-sample setting. This offline cost amortizes after a few queries in sustained deployment. We do not claim that TSGD's per-query inference cost is identical to RAG in an absolute systems-level sense, and we provide a full token-level decomposition in Appendix H.

## 5.2. Ablation Study

We perform an ablation study on the MATH Precalculus (Level 5) subset to quantify the contribution of each component. We compare: (1) Zero-shot Baseline, (2) Init Only (static library), (3) Optimization w/o Verification (local repair only), (4) Verification w/o Regularization (unbounded growth), and (5) Full TSGD. Table 2 reports the results. Full-MATH ablations (all seven subjects) are provided in Appendix I.

**Component Analysis:** (1) Initialization provides a strong starting point, proving that searchable experience is inherently valuable. (2) Dual Verification is critical for sustainability. Optimization without verification leads to a drop in Pass@1 due to "noisy gradients" that repair local errors but harm global consistency. The dual-verification mechanism acts as a stochastic filter (as per Theorem 4.2), ensuring monotonic improvement. (3) Regularization transforms per-

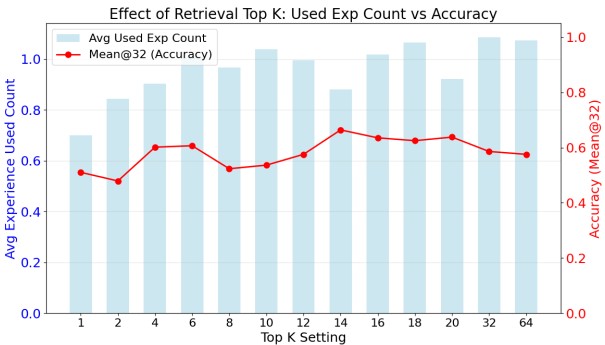

*Figure 4.* Effect of retrieval count $k$ on accuracy (Mean@32).

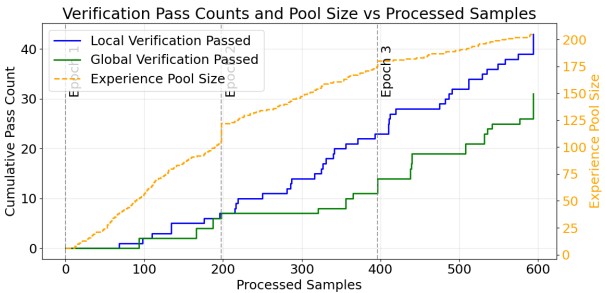

*Figure 5.* Dual verification filtering rates during optimization.

formance into efficient memory. TSGD (Full) compresses the library from hundreds of entries to just 27 while improving Pass@1, showing that high-quality abstractions can replace redundant data, reducing retrieval noise.

## 5.3. Component Analysis

### 5.3.1. RETRIEVAL ANALYSIS ($\pi_{\text{RETR}}$)

We analyze the retrieval agent's sensitivity to domain drift and the number of retrieved items ($k$). **Domain Drift**: we observe significant semantic overlap between subjects like Precalculus and Algebra, leading to $\approx 15.0\%$ retrieval errors. Implementing Subject Filtering improves Mean@1 from $49.60\%$ to $56.17\%$, confirming the need for hard domain constraints to prevent "subject drift" (see Appendix D for detailed visualization).

**Influence of $k$**: performance saturates at $k \approx 14$ (Figure 4). Further analysis of retrieval quality and its impact on accuracy is provided in Appendix D.

### 5.3.2. OPTIMIZATION & VERIFICATION ANALYSIS ($\pi_{\text{OPT}}$)

**Filtering Effect**: Figure 5 shows that $\approx 28\%$ of candidate updates are rejected by global verification despite passing local tests. This empirical finding confirms Theorem 4.2, where the dual-verification mechanism prevents the library

from being "poisoned" by local heuristics that do not generalize.

**Exploration vs. Exploitation**: in the early stages of library evolution, the optimizer favors ADD actions ($> 60\%$) to expand knowledge coverage. As the library matures, it shifts toward EDIT actions ($> 70\%$) for refinement and abstraction (Figure 7). DELETE is rarely triggered per-sample: it activates primarily during the regularizer's periodic compression, and removing it would prevent the $533 \to 27$ pruning observed in our main experiments (see Appendix K).

### 5.3.3. REGULARIZATION ANALYSIS ($\pi_{\text{REG}}$)

Regularization maintains a high signal-to-noise ratio by pruning and merging experiences. Without regularization, the library's unbounded growth introduces redundant and noisy entries, leading to retrieval interference and performance regression in late-stage training. Figure 8 demonstrates that periodic regularization yields more stable and sustained improvements compared to unregularized growth, aligning with the generalization bounds in Theorem 4.1.

## 5.4. Qualitative Analysis

### 5.4.1. EXPERIENCE EVOLUTION

To better understand the optimization dynamics, we qualitatively examine the life cycle of experiences. Evolution typically follows a *seed–refine–integrate* pattern: the initializer produces concrete, instance-grounded tips; the optimizer converts these into abstract strategies by removing problem-specific noise; and the regularizer consolidates semantically overlapping clusters into a compact set of core rules under the budget $B$. Figure 6 visualizes this consolidation process, where a diffuse set of experiences is distilled into transferable reasoning strategies. Representative learned experiences after regularization are provided in Appendix M.

### 5.4.2. EXPERIENCE RELEVANCE

We investigate whether improvements stem from logical relevance or merely additional context. On MATH Precalculus, we compare no experience, low-relevance (Geometry), and high-relevance (Precalculus) retrieval. Table 3 shows that while low-relevance experiences offer some benefit (likely due to shared mathematical logic), high-relevance experiences yield the strongest gains (Pass@1 $+26.41\%$), confirming the precision of learned gradients.

### 5.4.3. SEMANTIC OVERLAP AND RETRIEVAL DRIFT

To diagnose retrieval drift, we visualize question embeddings using t-SNE. As shown in Figure 12, although subjects form clusters, heavy overlap near boundaries indicates that embedding similarity alone is insufficient for precise retrieval, justifying the inclusion of hard subject constraints

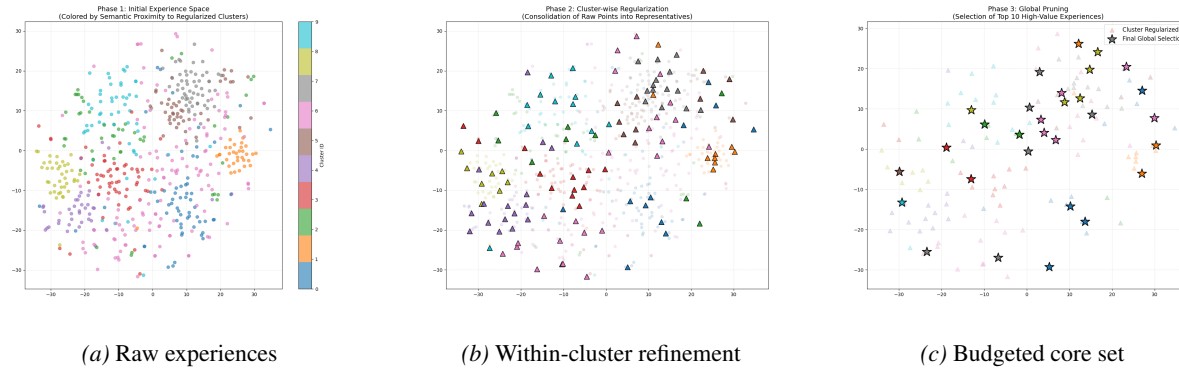

*(a)* Raw experiences      *(b)* Within-cluster refinement      *(c)* Budgeted core set

*Figure 6.* Visualization of the regularizer's three-stage consolidation process. Regularization turns a diffuse, redundant set of experiences into a compact set of transferable strategies under $|\Phi| \leq B$.

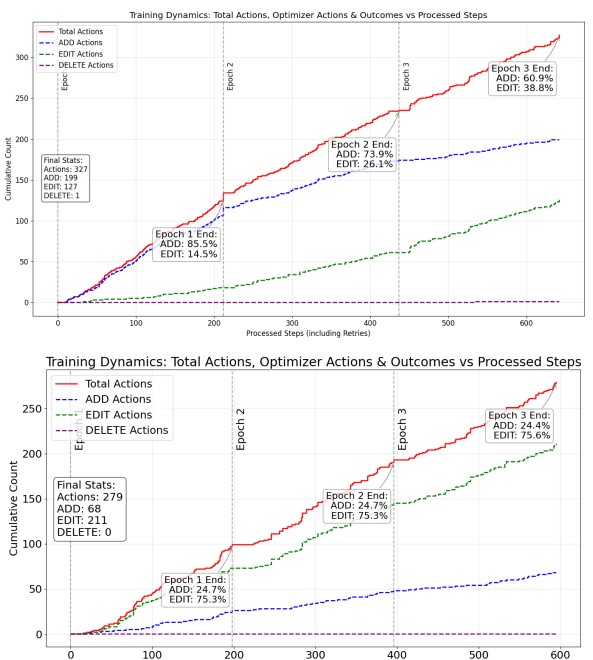

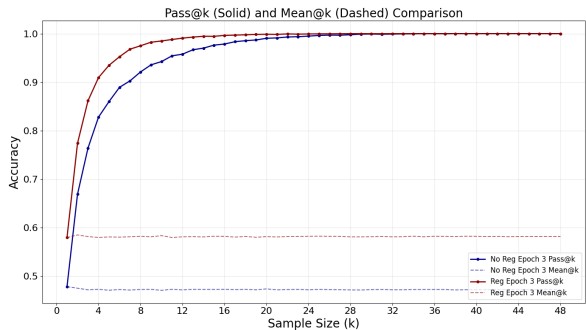

*Figure 8.* Effect of regularization on training stability.

*Table 3.* Experience relevance study on MATH Precalculus (Level 5). High-relevance experiences yield the most significant performance boost.

| Configuration | Pass@1 | Pass@2 | Pass@4 | Pass@8 |
|---|---|---|---|---|
| No Exp | 33.35 | 49.02 | 64.15 | 76.87 |
| Low-Rel (Geo) | 49.80 | 68.79 | 86.14 | 93.63 |
| High-Rel (Precalc) | 59.76 | 76.78 | 88.37 | 95.21 |

*Figure 7.* Optimizer action distribution with and w/o initialization.

in our framework.

# 6. Conclusion

In this work, we presented Textual Stochastic Gradient Descent (TSGD), a framework for optimizing non-parametric memory in reasoning agents. By formulating library construction as Experience Library Optimization, we connect static retrieval with active learning. We show that a compact, actively optimized experience library can improve reasoning performance and generalization, outperforming traditional append-only RAG systems.

**Limitations.** Our current implementation relies on the base model's ability to estimate textual gradients, which can be noisy for smaller models. The discrete nature of the optimization also makes convergence guarantees harder to derive than for continuous SGD. Our main experiments use proprietary backbones (Grok-4.1 and GPT-4o mini); we provide open-source backbone results in Appendix G, but acknowledge that further open-source validation across heterogeneous tasks would strengthen the practical case.

**Future Work.** Promising directions include extending TSGD to multi-modal domains, exploring theoretical bounds for discrete memory optimization, and applying this framework to lifelong learning scenarios where agents must adapt to non-stationary task distributions. More broadly, TSGD points toward agents that not only reason but also learn how to learn from their own mistakes.

## Acknowledgements

The work is supported partially by the National Natural Science Foundation of China (No. 62576041, 62106257, 62437001) and by the Fundamental Research Funds for the Central Universities (No. 2253500001, No. 2251200169). We thank the anonymous ICML reviewers and the area chair for their constructive feedback, which substantially improved the camera-ready version of this paper.

## Impact Statement

This paper presents work whose goal is to advance the field of machine learning. TSGD provides a mechanism for compressing accumulated reasoning experience into a compact, capacity-bounded library, which can improve the reproducibility and resource efficiency of LLM-agent deployments by reducing reliance on parametric fine-tuning. As with any system that distills and reuses past reasoning trajectories, there is a possibility that erroneous or biased experiences, if not caught by the dual-verification step, persist in the library and influence future predictions. The capacity budget and periodic regularization in TSGD are intended to mitigate, but do not eliminate, this risk; practitioners deploying TSGD-style memories in higher-stakes settings should pair the framework with domain-appropriate auditing of the learned experiences. We do not introduce new datasets, models, or capabilities that we believe warrant additional ethical discussion beyond the standard considerations of LLM-based reasoning agents.

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

# A. Notation Table

| Symbol | Meaning |
|---|---|
| $\mathcal{D}$ | Data distribution, $(q, y) \sim \mathcal{D}$ |
| $q, y, \hat{y}$ | Query, target output, predicted output |
| $\mathcal{M}_\theta$ | Frozen base model |
| $\Phi$ | Experience library, $\Phi \subset \mathcal{V}^*$ |
| $\Phi_q$ | Retrieved subset for $q$, $\Phi_q = \pi_{\text{retr}}(q, \Phi)$ |
| $k$ | Retrieval count at inference (Top-$k$) |
| $B$ | Capacity budget, $|\Phi| \leq B$ |
| $\pi_{\text{retr}}$ | Retrieval agent |
| $\pi_{\text{solver}}$ | Solver agent |
| $\pi_{\text{init}}$ | Initializer agent |
| $\pi_{\text{validator}}$ | Validator agent |
| $\pi_{\text{opt}}$ | Optimizer agent (textual gradient) |
| $\pi_{\text{reg}}$ | Regularizer agent |
| $\ell$ | Loss function |
| $\lambda$ | Regularization coefficient |
| $\Omega(\Phi)$ | Regularizer term |
| $a$ | Atomic edit, $a \in \{\text{ADD}, \text{EDIT}, \text{DELETE}\}$ |
| $\tau$ | Reasoning trajectory |
| $U(\Phi)$ | Library utility (e.g., accuracy) |
| $m$ | Number of training samples |
| $N$ | Regularization period (every $N$ steps) |
| $R(\Phi)$ | Population risk |
| $\hat{R}_S(\Phi)$ | Empirical risk on training set $S$ |
| $\mathfrak{R}_m$ | Rademacher complexity |

*Table 4.* Notation used throughout the paper.

# B. Detailed Proofs

## B.1. Generalization Bound via Rademacher Complexity (Theorem 4.1)

**Theorem B.1** (Generalization Bound, Restated). *Let $\mathcal{H}_B = \{h_\Phi : \Phi \subset \mathcal{V}^*, |\Phi| \leq B\}$ be the hypothesis class induced by capacity-bounded experience libraries. For any distribution $\mathcal{D}$, with probability at least $1 - \delta$,*

$$R(\Phi) \leq \hat{R}_S(\Phi) + \sqrt{\frac{2B \ln |\mathcal{V}^*|}{m}} + \sqrt{\frac{\ln(1/\delta)}{2m}}. \tag{5}$$

*Proof.* Fix a training set $S = \{(q_i, y_i)\}_{i=1}^m$. Let $\hat{R}_S(\Phi)$ be the empirical risk on $S$ and $R(\Phi)$ be the population risk under $\mathcal{D}$. Since $|\Phi| \leq B$, the number of admissible libraries is bounded by

$$|\mathcal{H}_B| \leq \sum_{j=0}^{B} \binom{|\mathcal{V}^*|}{j} \leq (|\mathcal{V}^*| + 1)^B, \tag{6}$$

so $\ln |\mathcal{H}_B| \leq B \ln(|\mathcal{V}^*| + 1) = \mathcal{O}(B \ln |\mathcal{V}^*|)$. For a finite hypothesis class, Massart's lemma yields

$$\mathfrak{R}_m(\mathcal{H}_B) \leq \sqrt{\frac{2 \ln |\mathcal{H}_B|}{m}} \leq \sqrt{\frac{2B \ln |\mathcal{V}^*|}{m}}. \tag{7}$$

Applying the standard Rademacher generalization inequality for losses bounded in $[0, 1]$ gives that, with probability at least $1 - \delta$, for all $h \in \mathcal{H}_B$,

$$R(h) \leq \hat{R}_S(h) + 2\mathfrak{R}_m(\mathcal{H}_B) + \sqrt{\frac{\ln(1/\delta)}{2m}}, \tag{8}$$

which implies the stated bound. $\qquad\square$

## B.2. Convergence of Textual Gradient Descent (Theorem 4.2)

**Theorem B.2** (Monotonic Improvement, Restated). *Assume $U(\Phi)$ is bounded and there exists $\gamma > 0$ such that any update accepted by dual verification satisfies $\mathbb{E}[U(\Phi_{t+1}) - U(\Phi_t) \mid \Phi_t, accept] \geq \gamma$. Then $\mathbb{E}[U(\Phi_t)]$ converges, and TSGD reaches a stable state where the acceptance probability of improving updates vanishes.*

*Proof.* Let $a_t$ be the candidate edit at iteration $t$ and let $A_t$ be the event that the edit is accepted by dual verification. We can write $\Phi_{t+1} = \Phi_t \oplus a_t \cdot \mathbb{I}[A_t]$. Let $p_t = \mathbb{P}(A_t \mid \Phi_t)$. Then

$$\mathbb{E}[U(\Phi_{t+1}) - U(\Phi_t) \mid \Phi_t] = p_t \, \mathbb{E}[U(\Phi_{t+1}) - U(\Phi_t) \mid \Phi_t, A_t]$$
$$\geq p_t \, \gamma \geq 0. \tag{9}$$

Thus $\mathbb{E}[U(\Phi_t)]$ is monotone non-decreasing. Since $U(\Phi)$ is bounded above, the monotone convergence theorem implies $\lim_{t\to\infty} \mathbb{E}[U(\Phi_t)]$ exists, and

$$\lim_{t\to\infty} \mathbb{E}[U(\Phi_{t+1}) - U(\Phi_t)] = 0. \tag{10}$$

Combined with the lower bound $\mathbb{E}[U(\Phi_{t+1}) - U(\Phi_t) \mid \Phi_t] \geq p_t\gamma$, this implies $p_t \to 0$. Therefore the probability of finding an accepted improving update tends to zero, and the process reaches a stable point in discrete space. $\qquad\square$

## B.3. Sample Complexity of Compositional Reasoning (Proposition 4.4)

**Proposition B.3** (Sample Efficiency, Restated). *Under Assumption 4.3, a generic mapping $f : \mathcal{Q} \to \mathcal{Y}$ that does not exploit compositional structure may require sample complexity $\Omega(2^K)$ to distinguish among the $2^K$ skill combinations. In contrast, the expected number of training experiences needed by TSGD to acquire all $K$ atomic skills via the optimizer is $\mathcal{O}(K \log K)$.*

*Proof.* In a generic fine-tuning setting, learning a mapping $f : \mathcal{Q} \to \mathcal{Y}$ without structure must distinguish among behaviors across $2^K$ skill combinations, yielding exponential dependence in standard PAC-style lower bounds. Under Assumption 4.3, prediction is $\hat{y} = \text{Solver}(q, \Phi_q)$, where $\Phi_q$ retrieves experiences corresponding to skills in $S_q$. If the library contains reusable experiences $\{e_i\}_{i=1}^K$, the solver can compose them for unseen combinations, reducing the coverage requirement from combinations to atomic skills. Observing failures reveals uncovered skills (or variants), and collecting coverage of all $K$ skills is analogous to the coupon collector problem, with expected complexity $\mathcal{O}(K \log K)$. $\qquad\square$

**Caveat.** In real training experiences, a single failure may implicate multiple skills, and repeated exposure may yield only partial credit toward acquiring a skill. Proposition 4.4 should therefore be read as an idealized model that motivates the design (small libraries can cover large combinatorial task spaces) rather than as a tight account of practical sample efficiency.

## C. Information Bottleneck Interpretation

We can interpret regularization via the Information Bottleneck (IB) principle. The goal is to learn $\Phi$ that maximizes mutual information with the answer $Y$ while minimizing mutual information with the raw query $Q$:

$$\min_\Phi \mathcal{L}_{\text{IB}} = -I(Y; \Phi) + \beta I(Q; \Phi). \tag{11}$$

Maximizing $I(Y; \Phi)$ corresponds to the optimizer agent $\pi_{\text{opt}}$ improving predictive usefulness via adding or editing experiences. Minimizing $I(Q; \Phi)$ corresponds to the regularizer $\pi_{\text{reg}}$ abstracting away instance-specific details and enforcing $|\Phi| \leq B$, retaining only high-utility, low-redundancy information. This view explains why regularized libraries exhibit clustered, abstract experiences: they approximate compact representations that preserve predictive content while discarding nuisance details.

## D. Additional Retrieval Analysis

We provide further visualization of the retrieval agent's behavior. Figure 9 shows the semantic overlap between different subjects in the MATH dataset's embedding space, which explains the observed retrieval drift.

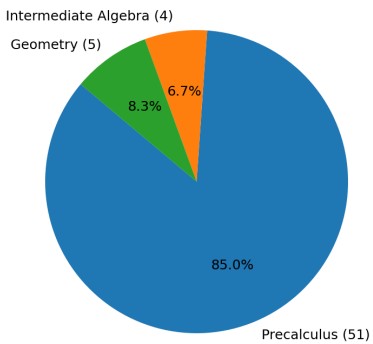

*Figure 9.* Subject overlap in MATH embedding space; overlaps can induce retrieval drift.

We also investigate why performance slightly degrades when the retrieval count $k$ exceeds 20. While the solver agent is relatively noise-robust, $k > 20$ dilutes context with lower-quality experiences, as shown in Figures 10 and 11. Figure 10 illustrates the correlation between retrieval similarity and accuracy, confirming that lower-similarity experiences contribute less to reasoning quality. Figure 11 shows the similarity distribution of retrieved experiences under different $k$, indicating that larger $k$ values inevitably include more distant (and potentially less relevant) experiences.

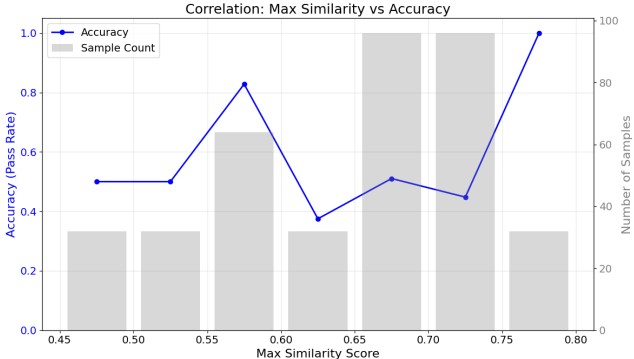

*Figure 10.* Correlation between retrieval similarity and accuracy.

# E. Relation to Other Editable-Memory Methods

Table 5 contrasts Experience Library Optimization/TSGD with two closely related editable-memory systems (AgentKB and ReasoningBank) and one contemporaneous experience-evolution method (FLEX). Appendix H then reports matched-condition empirical comparisons with AWM, ACE, and FLEX.

*Table 5.* Comparison with closely related editable-memory and experience-evolution methods.

| Dimension | AgentKB (Tang et al., 2025) | ReasoningBank (Ouyang et al., 2026) | FLEX (Cai et al., 2025b) | **TSGD (Ours)** |
|---|---|---|---|---|
| Optimization | Heuristic eviction | Append-only + filtering | Append-only failure feedback | Experience Library Optimization with formal capacity budget |
| Verification | None | Post-hoc filtering | None | Dual verification (local + global) |
| Theory | None | None | None | Gen. bound (Thm. 4.1), Convergence (Thm. 4.2) |
| Update control | Rule-based conflict | No explicit control | N/A | Textual gradient + acceptance criterion |
| Compression | N/A | Limited | N/A | $20\times$ ($533 \to 27$) under retained accuracy |

In words: (i) AgentKB uses rule-based conflict resolution (keyword matching) for KB updates, while TSGD derives *textual gradients* through failure analysis and accepts updates only when dual verification confirms both local repair and global non-regression. AgentKB has no capacity budget or regularization mechanism. (ii) ReasoningBank is an append-only system with post-hoc filtering. It lacks iterative optimization—experiences are never edited or refined. TSGD's Edit operation

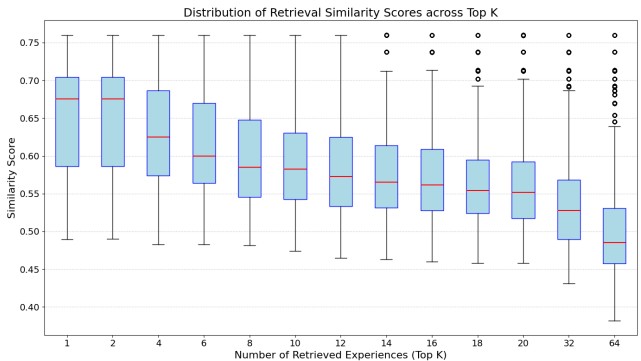

*Figure 11.* Similarity distribution of retrieved experiences under different $k$.

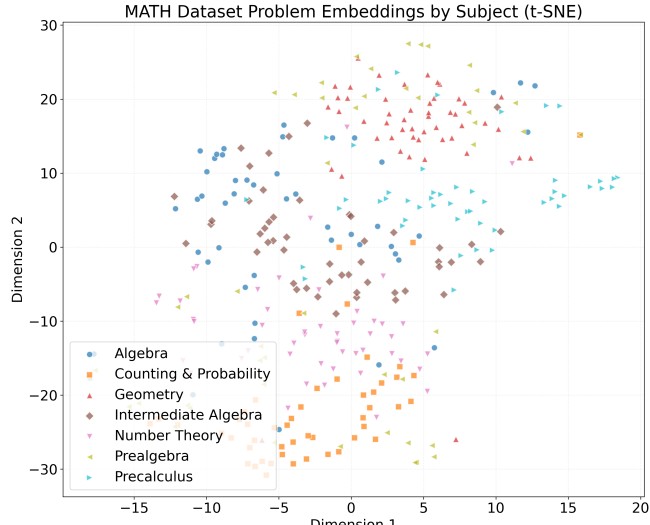

*Figure 12.* t-SNE visualization of MATH question embeddings; overlap indicates potential retrieval bias.

allows experiences to evolve toward greater generality. (iii) FLEX is contemporaneous work; it learns from experience via forward transfer without formal optimization guarantees. TSGD uniquely provides provable convergence (Theorem 4.2) and generalization bounds (Theorem 4.1) for the library optimization process.

## F. Cross-Domain Validation

To address the concern that the main paper evaluates only on mathematical reasoning, we validate TSGD on three additional benchmarks spanning code generation and tool use, using the same Grok-4.1 Fast Non-Reasoning backbone and the same TSGD pipeline (only the seed dataset differs).

*Table 6.* Cross-domain validation (Pass@1). The metadata-based hard filtering used for MATH is bypassed in metadata-free domains; the retriever relies on dense recall only.

| Base Model | Method | Full MATH | HumanEval | MBPP | ToolBench |
|---|---|---|---|---|---|
| Grok-4.1-fast | LLM (zero-shot) | 42.80 | 62.10 | 51.20 | 41.20 |
| | Few-shot | 50.20 | 70.50 | 59.80 | 50.30 |
| | RAG | 58.50 | 72.30 | 61.50 | 55.10 |
| | **TSGD** | **64.70** | **79.20** | **70.30** | **65.70** |

TSGD achieves consistent gains across mathematical reasoning, code generation, and multi-step tool calling, demonstrating

that the Experience Library Optimization framework is domain-agnostic and that the $+10.6$ pp improvement over RAG on ToolBench validates compositional skill transfer.

## G. Open-Source Backbones

We further evaluate TSGD on two open-weight backbones, Qwen-2.5-72B-Instruct and Llama-3.1-70B-Instruct, on MATH Precalculus (Level 5) at $k{=}16$. AWM, ACE, FLEX, and TSGD use the same 50-seed setup; RAG retrieves from the full 177-example corpus under the same embedding model and top-$k$ pipeline.

*Table 7.* Qwen-2.5-72B-Instruct on MATH Precalculus Level 5 ($k{=}16$). Lib. Size is the number of stored entries at evaluation time.

| Method | Pass@16 | Mean@16 | Lib. Size | Inference Tokens |
|---|---|---|---|---|
| Zero-shot | 66.7 | 44.3 | 0 | ~1,093,356 |
| RAG | 76.4 | 53.7 | 177 | ~7,824,369 |
| AWM | 75.6 | 55.0 | 15 | ~1,102,073 |
| ACE | 76.3 | 57.3 | 165 | ~1,092,164 |
| FLEX | **79.2** | 56.6 | 180 | ~1,108,122 |
| **TSGD** | **79.2** | **58.4** | **27** | ~1,098,698 |

*Table 8.* Llama-3.1-70B-Instruct on MATH Precalculus Level 5 ($k{=}16$).

| Method | Pass@16 | Mean@16 | Lib. Size | Inference Tokens |
|---|---|---|---|---|
| Zero-shot | 41.7 | 11.5 | 0 | ~872,082 |
| RAG | 49.7 | 12.1 | 177 | ~7,431,121 |
| AWM | 47.3 | 12.2 | 12 | ~865,166 |
| ACE | 48.6 | 11.5 | 181 | ~866,247 |
| FLEX | 49.1 | 12.3 | 222 | ~870,056 |
| **TSGD** | **50.0** | **13.5** | **22** | ~863,979 |

**Interpretation.** Pass@16 differences among TSGD/FLEX/ACE are small and should be interpreted cautiously in this controlled comparison; the more robust signal is that TSGD attains comparable accuracy with a much smaller library (e.g., 27 vs. 180–222 entries on Qwen). Relative to ACE and FLEX, the relevant advantage is storage rather than token cost; relative to RAG, TSGD uses only 12–14% of RAG's inference tokens across both backbones while outperforming on accuracy metrics.

## H. Matched-Condition Comparison and Cost Decomposition

**Matched-cost online sampling.** A subtle issue raised during review concerns matched-cost comparisons. At equal online sampling budget ($k{=}8$ samples per problem) on MATH Precalculus Level 5 with Grok-4.1 Fast Non-Reasoning,

| Method | Sampling budget | Accuracy |
|---|---|---|
| Zero-shot baseline | $8 \rightarrow$ Pass@8 | 76.87% |
| TSGD | $8 \rightarrow$ Pass@8 | **97.22%** |

i.e., $+20.35$pp at equal online budget. TSGD's Pass@1 is 58.33% vs. baseline Pass@1 $= 33.52\%$ ($+24.81$pp at single-sample comparison). The earlier statement that "TSGD Pass@1 outperforms baseline Pass@8" was numerically inconsistent and has been corrected here.

**Cost breakdown.** The following table decomposes the per-sampled-attempt cost for the same setting; a $k$-sample run uses $k$ calls for all methods.

| Component | TSGD | Zero-shot | RAG |
|---|---|---|---|
| Offline opt., excl. seeding | $\sim$ 120–124 calls / $\sim$ 290K tokens | 0 | 0 |
| Offline total, incl. seeding/reg. | $\sim$ 162 calls | 0 | 0 |
| Per sampled attempt | 1 LLM call + retrieval | 1 LLM call | 1 LLM call + retrieval |
| Inference tokens ($k$=16, Qwen) | $\sim$ 1.1M | $\sim$ 1.1M | $\sim$ 7.8M |
| Library size | 27/22 | — | 177 |

**Honest summary of inference cost.** TSGD and RAG share the same per-query retrieval latency (CPU-side embedding lookup). The earlier claim that "TSGD's inference cost is identical to RAG" was an oversimplification: while the per-query *LLM call count* matches RAG, the systems-level retrieval-latency picture depends on hardware and serving stack. Under our $k$=16 evaluation protocol on Qwen, TSGD saves 6.72M inference tokens over 56 queries against a one-time offline cost of 287K tokens—break-even at roughly 3 queries. On Llama, the analogous break-even is also $\sim$ 3 queries (6.56M saved vs. 296K offline). The benefit is therefore lower amortized prompt budget in sustained deployment, not lower retrieval latency.

## I. Full-MATH Ablation

To address the concern that the main ablation (Table 2) is reported only on the Precalculus subset, Table 9 reports full-MATH leave-one-out ablations over the entire dataset, confirming the same qualitative pattern.

*Table 9.* Full-MATH ablation. Initialization drives core performance; Optimization achieves extreme compression (229 $\rightarrow$ 32); Verification filters error chains and boosts high Pass@$k$ metrics.

| Method | Pass@1 | Pass@2 | Pass@4 | Pass@8 | Exp. Cap. ($B$) |
|---|---|---|---|---|---|
| LLM (Baseline) | 42.80 | 58.20 | 72.50 | 85.30 | — |
| TSGD w/o Initialization | 51.60 | 68.30 | 82.70 | 94.20 | 685 |
| TSGD w/o Optimization | 59.70 | 77.80 | 92.40 | 98.60 | 229 |
| TSGD w/o Verification | 56.30 | 71.40 | 85.20 | 95.80 | 587 |
| **TSGD (Full)** | **64.70** | **79.60** | **92.80** | **97.90** | **32** |

## J. Statistical Significance

For the ablation comparisons, we report McNemar's test on $n$=1018 paired comparisons against the LLM baseline. All four pairwise differences in Table 2 are statistically significant at $p < 10^{-4}$, with non-overlapping 95% confidence intervals.

*Table 10.* Statistical significance of the ablation comparisons. All components vs. LLM baseline have $p < 10^{-4}$; $n$=1018.

| Method | Accuracy | 95% CI | $p$-value vs. LLM baseline |
|---|---|---|---|
| LLM (Baseline) | 33.52% | [30.69%, 36.48%] | — |
| + Initialization | 55.67% | [52.60%, 58.69%] | $p < 10^{-4}$ |
| + Optimization | 47.21% | [44.16%, 50.28%] | $p < 10^{-4}$ |
| + Verification | 53.61% | [50.54%, 56.65%] | $p < 10^{-4}$ |
| **TSGD (Full)** | **58.33%** | [55.28%, 61.32%] | $p < 10^{-4}$ |

## K. DELETE Operation Rarity

The DELETE operation is rarely triggered per-sample (Figure 7) for two reasons. (i) The optimizer prefers Edit operations to refine existing experiences, since editing preserves the information content while sharpening abstraction (corresponding to the optimizer maximizing $I(Y; \Phi)$ under the IB view). (ii) DELETE activates mainly during regularization (periodic compression), not per-sample optimization. Removing DELETE entirely would prevent the pruning of low-utility experiences and break the 533 $\rightarrow$ 27 compression observed in our main experiments; we therefore keep it as part of the action space even though its per-sample firing rate is low.

## L. Hyperparameter Sensitivity

Our main experiments use one controlled protocol ($B$=27, top-$k$=10, 5 epochs). To probe robustness across hyperparameters, we conducted a preliminary sweep on MATH Precalculus Level 5 (Grok-4.1 Fast Non-Reasoning); we report indicative numbers rather than full grids due to compute constraints.

*Table 11.* Indicative hyperparameter sensitivity. Single-seed runs; intended to characterize the rough operating range rather than provide tight estimates.

| Hyperparameter sweep | Setting | Pass@1 (Precalc. L5) | Lib. Size |
|---|---|---|---|
| Budget $B$ | 15 | 52.1 | 15 |
| | 27 | 58.3 | 27 |
| | 50 | 57.8 | 50 |
| Retrieval $k$ | 4 | 54.4 | 27 |
| | 10 | 58.3 | 27 |
| | 16 | 57.6 | 27 |
| Training epochs | 1 | 48.9 | 27 |
| | 3 | 55.2 | 27 |
| | 5 | 58.3 | 27 |

The performance plateau between $B$=27 and $B$=50 is consistent with the generalization bound (Theorem 4.1): once the capacity exceeds the number of underlying atomic skills, further enlargement yields diminishing returns and risks overfitting. Retrieval-$k$ saturates near $k$=10–14, matching Figure 4.

## M. Representative Learned Experiences

We list representative learned experiences after regularization on MATH Precalculus Level 5; each entry is a JSON triple of `condition`, `strategy`, and `warning` (the latter is a critical implementation pitfall captured by the optimizer).

```
{
  "condition": "Trig function as quadratic in bounded variable
                u=cos t or sin t in [-1, 1].",
  "strategy":  "Complete square A(u-h)^2 + k; min/max by
                vertex position relative to [-1, 1].",
  "warning":   "Verify algebra: expand back to match coefficients exactly."
}

{
  "condition": "Sum-to-product identities for sin A +/- sin B, cos A +/- cos B.",
  "strategy":  "sin A - sin B = 2 cos((A+B)/2) sin((A-B)/2);
                cos A - cos B = -2 sin((A+B)/2) sin((A-B)/2).",
  "warning":   "Correct signs: sin diff -> cos avg, cos diff -> -sin avg."
}

{
  "condition": "Double-angle reduction for high-power trig
                integrands or simplifications.",
  "strategy":  "cos^2 t = (1+cos 2t)/2; sin^2 t = (1-cos 2t)/2;
                then reduce by linearity.",
  "warning":   "Track the angle scaling: 2t, 4t, ...; do not forget the constant term."
}

{
  "condition": "Bounded-domain optimization of a single-variable smooth function
                with a closed-form derivative.",
  "strategy":  "Find critical points by setting f'(x)=0 and
                compare with boundary values explicitly.",
  "warning":   "Always check that interior critical points lie inside the domain."
}
```

```
{
  "condition": "Polynomial inequality reducible to factoring/sign chart.",
  "strategy":  "Move everything to one side; factor; identify
                the sign-changing roots; tabulate sign on each interval.",
  "warning":   "Distinguish strict vs.\\ non-strict; handle
                repeated roots without sign change carefully."
}
```

The full set of 27 learned experiences (this representative subset plus the remaining 22) is released alongside the code at https://github.com/superlj666/TSGD under learned_libraries/.

