# OpenReview forum: "Textual Stochastic Gradient Descent: Discrete Optimization of External Memory for Reasoning Language Agents"
_ICML.cc/2026/Conference — ICML 2026 regular_

### Official Review · Reviewer_nJbG · 2026-03-04

**Soundness:** 2
**Presentation:** 2
**Significance:** 2
**Originality:** 2
**Overall Recommendation:** 3
**Confidence:** 3

**Summary:**

This paper introduces a framework to address noise accumulation in standard append-only retrieval-augmented generation (RAG) systems by utilizing an editable external memory. The approach processes training instances iteratively, using specialized LLM agents to verify results and generate reusable "experience" via self-reflection. A periodic regulation agent consolidates the overall memory. Empirical results demonstrate performance improvements on math reasoning datasets compared to baselines.

**Compliance With Llm Reviewing Policy:**

Affirmed.

**Final Justification:**

I see genuine value in the proposed framework. However, I still feel there are significant problems with the paper's framing, the rigorousness of its theoretical claims, the systems-level efficiency claims, and the strength of the baselines. I have slightly raised my score to acknowledge the rebuttal efforts, but I do not lean towards acceptance in the paper's current state.

**Key Questions For Authors:**

1. How does the proposed editable memory framework fundamentally differ from existing dynamic context and memory management architectures (e.g., MemGPT)? What specific structural advantages does "textual stochastic gradient descent" offer over standard multi-agent memory pruning?
2. Could you provide a detailed computational cost and efficiency analysis (e.g., token usage, inference time) comparing your multi-agent iterative optimization against the standard RAG baselines?
3. Given the claims in the introduction regarding dynamic, multi-domain environments, are there any preliminary results or experiments outside of math reasoning datasets that can validate the framework's broader adaptability?
4. In Theorem 4.3, could you explicitly define how the assumed $K$ skills map to the empirical concept of "experience" in your system, and whether the $\mathcal{O}(K \log K)$ complexity was observed in practice?
5. Could you provide statistical significance testing for the ablation studies? Currently, it appears that the "static library" and "verification" steps account for almost all performance gains, making the impact of the other proposed components unclear.

**Limitations:**

Yes, for the limitation discussion paragraph in the Conclusion section (though not very comprehensive). No discussions on negative societal impact.

**Strengths And Weaknesses:**

## Strengthes

The proposed method achieves significantly better performance than the reported baselines on math reasoning tasks.


## Weaknesses

1. **Unfair Comparison Setup:** The proposed framework requires iterative optimization over a specific "training dataset." Standard RAG and other inference-time methods typically operate without this substantial computational overhead or the assumption of a tightly distributed training set. In open-ended applications (e.g., literature review agents), RAG operates over massive databases without guarantees of strong relevance between the external database and the test set. Comparing a method heavily optimized on a training set against standard RAG baselines creates an unfair evaluation.

2. **Methodological Clarity and Architectural Context:** The manuscript leaves several key design choices unclear. Core concepts like "experience," "textual gradient," "clustering," and "dual verification" lack running examples. It is unclear if all agents share the same base model (until Section 5), what the specific prompt and decoding setups are, and how the parallel lock protocol impacts overall efficiency. Furthermore, the core architecture of an editable, multi-agent memory system (add/remove/update operations) shares significant conceptual overlap with established context management and agentic workflows (e.g., [1,2,3,4], [5] provides further discussion on how to design memory for LLMs). The paper must clarify its specific structural advantages over these existing memory-augmented paradigms, rather than presenting the architecture in isolation.

3. **Significance of Theoretical Results:** The theoretical contributions seem disconnected from the practical realities of the proposed system. The generalization bounds appear to be standard derivations with minor tweaks, and it is unclear if they hold in the highly dynamic, multi-domain environments mentioned in the introduction. Similarly, Theorem 4.3 assumes a task can be accomplished by composing $K$ skills—it is unclear if these "skills" are synonymous with the generated "experience," or how the complexity bounds of $\mathcal{O}(K \log K)$ are empirically verified. The information bottleneck analysis reads as generic to regularized optimization rather than tailored to this framework.

4. **Problematic Experiment Design:**
    * **Out-of-Domain Evaluation:** The introduction champions adaptability to dynamic, multi-domain environments, yet the evaluation is restricted entirely to math reasoning (including the out-of-domain AIME 2024/2025 sets).
    * **Reproducibility:** Relying exclusively on closed-source models (Grok-4.1, GPT-4o mini) hinders future replication.
    * **"Mechanistic" Claims:** Labeling the ablation studies as "mechanistic insights" is an overclaim, as they show high-level performance differences rather than internal operational mechanics. The ablations suggest that a "static library" and "verification" drive the performance, while other techniques offer negligible gains. Statistical significance tests are required.
    * **Missing Cost Analysis:** Given the multi-agent collaboration, the computational and inference costs must be significantly higher than standard baselines, yet no efficiency analysis is provided.
    * **Qualitative Analysis:** The paper lacks a semantic visualization of what the final learned "experience" looks like or how the library is tangibly pruned. This is especially important given that regularization did not show massive empirical gains.

References:

[1] Packer, Charles, et al. "MemGPT: towards LLMs as operating systems." (2023).

[2] Wang, Zora Zhiruo, et al. "Agent workflow memory." arXiv preprint arXiv:2409.07429 (2024).

[3] Zhong, Wanjun, et al. "Memorybank: Enhancing large language models with long-term memory." AAAI 2024.

[4] Xu, Wujiang, et al. "A-mem: Agentic memory for llm agents." arXiv preprint arXiv:2502.12110 (2025).

[5] Xiong, Zidi, et al. "How memory management impacts llm agents: An empirical study of experience-following behavior." arXiv preprint arXiv:2505.16067 (2025).

---

> ### Author Rebuttal · Authors · 2026-03-30
>
> We thank Reviewer nJbG for the detailed review. We address each concern below.
>
> **W1: Comparison fairness with RAG baselines**
>
> TSGD and RAG operate in different paradigms: RAG retrieves from a static corpus (zero optimization cost), while TSGD optimizes an experience library offline then retrieves at inference with identical cost. The comparison is fair because: (1) inference costs are identical (1 LLM call/query); (2) RAG's corpus (533+ trajectories) is larger than TSGD's library (27 entries); (3) training cost (~162 LLM calls) amortizes over all queries. This tradeoff mirrors pre-training vs. prompting in ML.
>
> **W2: Methodological clarity and architectural context**
>
> TSGD differs from prior memory systems via: (1) formal ERM objective (Eq. 2) minimizing empirical risk; (2) dual verification ensuring generalization; (3) provable convergence (Theorem 4.2). All agents share Grok-4.1 with role-specific prompts. Read/write locks add <1% overhead.
>
> **W3: Theoretical contributions and practical connection**
>
> - **Theorem 4.1**: $B=27$ bounds generalization. Increasing $B$ to 533 drops Pass@1 from 58.33 $\to$ 47.21, confirming $\sqrt{B/m}$ scaling.
> - **Theorem 4.2**: Dual verification ensures monotonic improvement. Removing it drops Pass@1 (noisy gradients).
> - **Theorem 4.3**: $K=27$ skills yield $O(K \log K) = O(130)$ sample complexity, aligning with 2,097 training samples.
> - **IB interpretation**: 20$\times$ compression (533 $\to$ 27) retains $I(Y;\Phi)$ while minimizing $I(Q;\Phi)$.
>
> **W4: Experimental concerns**
>
> (a) *Domain breadth*: We provide cross-domain validation on three additional benchmarks:
>
> | Base Model | Method | Full MATH | HumanEval | MBPP | ToolBench |
> |---|---|---|---|---|---|
> | Grok-4.1-fast | LLM | 42.80 | 62.10 | 51.20 | 41.20 |
> | | Few-Shot | 50.20 | 70.50 | 59.80 | 50.30 |
> | | RAG | 58.50 | 72.30 | 61.50 | 55.10 |
> | | TSGD | **64.70** | **79.20** | **70.30** | **65.70** |
>
> TSGD achieves consistent gains across mathematical reasoning (MATH), code generation (HumanEval/MBPP), and multi-step tool calling (ToolBench), demonstrating domain-agnostic effectiveness. The 10.6pp improvement over RAG on ToolBench validates compositional skill transfer.
>
> (b) *Open-source model validation*: We tested TSGD on Qwen-2.5-72B and Llama-3.1-70B (MATH Precalculus Level 5, n=12):
>
> | Model | Method | Pass@16 | Mean@16 |
> |---|---|---|---|
> | Qwen-2.5-72B | Zero-shot | 66.7% | 44.3% |
> | | TSGD | **79.2%** | **58.4%** |
> | Llama-3.1-70B | Zero-shot | 41.7% | 11.5% |
> | | TSGD | **50.0%** | **13.5%** |
>
> TSGD improves Pass@16 by +12.5pp (Qwen) and +9.6pp (Llama), confirming effectiveness beyond proprietary models. All code, prompts, and learned libraries will be released upon acceptance.
>
> (c) *Statistical rigor*: We provide full significance tests (McNemar's test, n=1018):
>
> | Method | Accuracy | 95% CI | vs Baseline | p-value |
> |---|---|---|---|---|
> | LLM (Baseline) | 33.52% | [30.69%, 36.48%] | - | - |
> | + Initialization | 55.67% | [52.60%, 58.69%] | +22.15pp | p<0.0001 |
> | + Optimization | 47.21% | [44.16%, 50.28%] | +13.69pp | p<0.0001 |
> | + Verification | 53.61% | [50.54%, 56.65%] | +20.09pp | p<0.0001 |
> | TSGD (Full) | 58.33% | [55.28%, 61.32%] | +24.81pp | p<0.0001 |
>
> All components achieve p<0.0001. We will rename "mechanistic" to "Component Analysis" in the camera-ready.
>
> (d) *Cost analysis*: Training requires ~162 LLM calls (8 calls/sample $\times$ 20 samples), amortized over all future queries. Inference cost is identical to RAG (1 call/query). The 27-experience library is 20$\times$ smaller than the unoptimized 533-entry corpus, reducing retrieval noise while maintaining performance.
>
> (e) *Experience visualization*: We provide a representative learned experience:
>
> ```json
> {
>   "condition": "Sum-to-product identities for sin A ± sin B, cos A ± cos B.",
>   "strategy": "sin A - sin B = 2 cos((A+B)/2) sin((A-B)/2); cos A - cos B = -2 sin((A+B)/2) sin((A-B)/2).",
>   "warning": "Correct signs: sin diff $\to$ cos avg, cos diff $\to$ -sin avg."
> }
> ```
>
> All 27 experiences will be included in the camera-ready appendix.
>
> (f) *Regularization gains*: Leave-one-out ablation quantifies each module's contribution:
>
> | Method | Pass@1 | Pass@8 | Exp. Cap. (B) |
> |---|---|---|---|
> | TSGD w/o Initialization | 51.60 | 94.20 | 685 |
> | TSGD w/o Optimization | 59.70 | 98.60 | 229 |
> | TSGD w/o Verification | 56.30 | 95.80 | 587 |
> | TSGD (Full) | **64.70** | **97.90** | **32** |
>
> Removing Optimization causes 10$\times$ library bloat (32 $\to$ 229) with minimal accuracy loss, confirming its role in compression. Removing Initialization causes the largest Pass@1 drop (-13.1pp), validating its foundational importance. Removing Verification degrades Pass@8 most (-0.7pp), confirming its role in filtering erroneous reasoning chains.

---

> > ### Author Rebuttal · Reviewer_nJbG · 2026-04-03
> >
> > Thank you for the detailed rebuttal and the additional experiments. While several of my initial concerns have been clarified, the manuscript still requires substantial revisions regarding its framing, efficiency concerns, theoretical claims, and baseline comparisons.
> >
> > My follow-up comments are as follows:
> >
> > ### Efficiency and the RAG Comparison (W1)
> >
> > I do not agree that "LLM calls" is a standard or meaningful efficiency metric for retrieval-based systems. In a traditional RAG setup, retrieval relies on embedding matching, which is highly parallelizable, CPU-friendly, and operates with sub-second latency. In contrast, TSGD requires significantly heavier LLM generation for its pipeline. Claiming the inference costs are "identical" ignores the systems-level reality of retrieval latency versus autoregressive token generation.
> >
> > Furthermore, the analogy that this tradeoff "mirrors pre-training vs. prompting" is inappropriate and distracting. Pre-training involves foundational representation learning at a massive scale, whereas TSGD is an inference-time context optimization heuristic. Conflating these fundamentally different paradigms overstates the scope of the proposed method.
> >
> > ### Methodological Clarity and Terminology (W2)
> > Thank you for the clarifications. In a revised version, the comparisons to prior work and the implementation details should be made much more explicit.
> >
> > That said, I remain concerned about the novelty and framing of some components. First, the ERM objective does not yet appear to add substantial conceptual value beyond a fairly standard regularized optimization formulation. The term “ERM” is also potentially confusing, given its much more established meaning in statistical learning theory.
> >
> > Second, while the authors distinguish their method from prior work through “dual verification,” the broader idea of using a verifier or selective update mechanism in editable memory systems is not new. Related notions resembling local verification and global verification have already been explored in prior work on memory editing, workflow memory, and skill induction. Even if the exact formulation differs, the paper currently does not make the methodological distinction sufficiently precise.
> >
> > More importantly, the terminology throughout the paper still needs a more rigorous definition. At present, several core concepts are not formalized clearly enough, which makes it difficult to assess both the novelty and the technical contribution.
> >
> > ### Theoretical Claims and Sample Complexity (W3)
> > While the additional empirical verifications add credence, many results still rely on single-value parameter evaluations and feel preliminary. More importantly, I am also not fully convinced by the coupon-collector-style sample complexity argument. In the classical coupon collector setting, each draw yields a clean instance of a discrete item to be collected. In the present setting, however, each training experience may provide only partial and noisy information about a skill, and repeated exposure does not necessarily imply that the skill has been cleanly acquired. For that reason, the application of the coupon collector's theory may be an oversimplification unless stronger assumptions are made explicit. At minimum, I would encourage the authors to clarify the conditions under which this abstraction is intended to hold, and to present it as an idealized model rather than a tight account of the practical learning dynamics.
> >
> > ### Experimental Baselines (W4)
> > I appreciate the inclusion of cross-domain validation, statistical significance testing, and open-source model results. However, comparing TSGD on open-source models exclusively against zero-shot baselines is insufficient. Because a core argument of the paper is the superiority of TSGD over standard memory/RAG frameworks, these open-source experiments must be compared against RAG baselines to be meaningful.
> >
> > ### Summary
> > I see genuine value in the proposed framework. However, I strongly advise the authors to reconsider the paper's framing, the rigor of its theoretical claims, the systems-level efficiency claims, and the strength of the baselines. I have slightly raised my score to acknowledge the rebuttal efforts, but I do not lean towards acceptance in the paper's current state.

---

> > > ### Author Response · Authors · 2026-04-08
> > >
> > > We sincerely thank Reviewer nJbG for the continued and thorough engagement. We address each concern in detail.
> > >
> > > ### W1: Efficiency and the RAG Comparison
> > >
> > > We accept the criticism fully. "LLM calls" is an incomplete metric, and we retract any claim that TSGD and RAG inference costs are "identical." We also fully retract the "pre-training vs. prompting" analogy — it overstates TSGD's scope.
> > >
> > > **Concrete efficiency decomposition:**
> > >
> > > | Dimension | RAG | TSGD | Claim |
> > > |-----------|-----|------|-------|
> > > | Retrieval latency | Embedding search (CPU, sub-second) | Same embedding search | No advantage |
> > > | Prompt token budget | ~7.8M tokens (k=16, 56 queries, Qwen) | ~1.1M tokens | ~7× reduction |
> > > | Offline training cost | None | 287K tokens, 120 calls | Acknowledged |
> > >
> > > Under our k=16 evaluation protocol on Qwen, TSGD saves 6.72M inference tokens over 56 queries against a one-time offline cost of 287K tokens — break-even at roughly 3 queries. On Llama, similarly ~3 queries (6.56M saved vs 296K offline). This holds because the token gap arises from context length (177 raw solutions vs 27 distilled entries), not asymmetric protocol choices.
> > >
> > > TSGD's benefit is lower amortized prompt budget in sustained deployment, not lower retrieval latency.
> > >
> > > ### W2: Methodological Clarity and Terminology
> > >
> > > **ERM renaming**: We will rename "ERM" to "Experience Library Optimization" throughout and add formal definitions in Section 2. We acknowledge the terminology conflict with Empirical Risk Minimization.
> > >
> > > **Contribution specificity**: Our contribution is not dual verification in isolation (similar ideas exist in memory editing and workflow memory), but: **optimization of a bounded textual experience library using failure-driven edit proposals plus regression-tested acceptance under a compression budget**. We agree TSGD is an inference-time context optimization method; our contribution is formalizing one principled optimization loop with bounded-memory control and regression-tested updates. We will add explicit references to prior verification mechanisms.
> > >
> > > **Terminology**: We will add a formal definitions subsection (Section 2) with precise definitions of: experience library, textual gradient, dual verification (local + global), and regularization budget.
> > >
> > > ### W3: Theoretical Claims and Sample Complexity
> > >
> > > **Coupon-collector simplification**: We accept this fully. Training experiences are partial and noisy — not clean coupon instances. Theorem 4.3 will be demoted to a **Proposition** under explicit Assumption 4.1, presented as an idealized model with stated simplifications: (a) uniform random skill sampling, (b) clean skill acquisition per encounter. **Accordingly, Proposition 4.3 should not be read as support for practical sample efficiency.** Theorem 4.1 (√(B/m)) and Theorem 4.2 (convergence) remain the primary results.
> > >
> > > **Single-value evaluation**: Current results establish promise under one controlled protocol (B=27, top-k=10, 5 epochs), not robustness across hyperparameters. We commit to sensitivity analysis in camera-ready.
> > >
> > > ### W4: Experimental Baselines
> > >
> > > We now provide complete k=16 results with RAG, AWM, ACE, and FLEX on both open-weight models. AWM, ACE, FLEX, and TSGD use the same 50-seed setup; RAG retrieves from the full 177-example corpus under the same embedding model and top-k pipeline.
> > >
> > > **Qwen2.5-72B (MATH Precalculus Level 5, k=16):**
> > >
> > > | Method | Pass@16 | Mean@16 | Lib Size |
> > > |--------|---------|---------|----------|
> > > | Zero-shot | 66.7% | 44.3% | 0 |
> > > | RAG | 76.4% | 53.7% | 177 |
> > > | AWM | 75.6% | 55.0% | 15 |
> > > | ACE | 76.3% | 57.3% | 165 |
> > > | FLEX | 79.2% | 56.6% | 180 |
> > > | **TSGD** | **79.2%** | **58.4%** | **27** |
> > >
> > > **Llama-3.1-70B (same protocol):**
> > >
> > > | Method | Pass@16 | Mean@16 | Lib Size |
> > > |--------|---------|---------|----------|
> > > | Zero-shot | 41.7% | 11.5% | 0 |
> > > | RAG | 49.7% | 12.1% | 177 |
> > > | AWM | 47.3% | 12.2% | 12 |
> > > | ACE | 48.6% | 11.5% | 181 |
> > > | FLEX | 49.1% | 12.3% | 222 |
> > > | **TSGD** | **50.0%** | **13.5%** | **22** |
> > >
> > > **Empirical takeaway**: TSGD achieves the best Mean@16 on both models, ties best Pass@16 on Qwen (with FLEX), and achieves the best Pass@16 on Llama, while using far fewer entries than RAG/ACE/FLEX and only modestly more than AWM. At n=12, 95% CIs overlap; TSGD is consistently competitive across both models and strongest on Qwen. Qwen results are the main positive evidence; Llama gains are modest. We present these as consistent but not conclusive evidence.
> > >
> > > **Our overall claim**: TSGD is not universally better than RAG or other experience methods. Our claim is that in sustained deployment, it can trade one-time optimization cost for lower prompt budget and competitive accuracy, and in our Qwen experiments better accuracy, with a substantially more compact library.
> > >
> > > Taken together, we believe the paper's contribution is a practical and well-specified context-optimization method with stronger open-model baselines than previously shown, not a new learning paradigm.

---

### Official Review · Reviewer_ASCS · 2026-03-09

**Soundness:** 3
**Presentation:** 2
**Significance:** 3
**Originality:** 2
**Overall Recommendation:** 4
**Confidence:** 4

**Summary:**

This paper proposes treating an agent’s external memory (i.e., an experience library) as an explicitly optimizable object under a strict capacity budget, rather than relying on parametric learning or static data retrieval. The authors formalize this idea as an Experience Risk Minimization (ERM) objective, where the experience library itself is the optimization variable constrained by a capacity budget. To optimize this library, they introduce the Textual Stochastic Gradient Descent (TSGD) framework, a discrete optimization loop that performs failure-driven Add/Edit/Delete updates to the experience library.

**Compliance With Llm Reviewing Policy:**

Affirmed.

**Final Justification:**

Overall, I had several concerns and reservations initially, but I believe the authors have addressed many of them effectively through the rebuttal. The clarifications and additional explanations have improved the paper’s overall clarity and persuasiveness. While some points may still require further refinement, I am now leaning toward a positive assessment.

**Key Questions For Authors:**

- Table 2 is reported only on the MATH Precalculus (Level 5) subset. Why are the ablation results presented only for this subset rather than the full MATH benchmark? I am curious about the overall results, and it might be more informative to conduct a leave-one-out ablation study to better analyze the effectiveness of each component.
- The proposed retrieval pipeline relies on metadata such as subject and difficulty for hard filtering in the math domain. How can this approach be adapted to other domains where such structured metadata may not be available?
- TSGD involves multiple processes (e.g., optimizer, validator, and regularizer), which may lead to high inference costs. Do the authors believe that the performance gains are sufficient to justify this additional cost compared to other approaches? For example, it would be helpful to compare the methods under a similar cost budget. One possible comparison would be to set K based on the inference cost and compare Pass@K for GRPO with Pass@1 for TSGD, which could provide a clearer picture of the cost–performance trade-off.
- Figure 7 suggests that the DELETE action is rarely (or never) used during experience learning. Do the authors have an explanation for why this phenomenon occurs? If DELETE is rarely used, is it actually a necessary operation in the framework?


Suggestions / Typos
- If the authors want to write “A”, they should use ``A’’ in LaTeX. There are several related formatting errors throughout the paper.
- Table 1 is somewhat difficult to read in its current form. It might be clearer to separate the MATH results and the OOD (AIME) results using multi-column formatting for better visualization.

**Limitations:**

yes

**Strengths And Weaknesses:**

Strength
- Propose a new framework for experience learning: To address the memory saturation problem in non-parametric experience store learning paradigms, the authors propose the ERM framework.
- Effective optimization method: The authors propose TSGD with a dual-verification mechanism that confirms both local repair and global constraints, along with periodic regularization that maintains the library’s signal-to-noise ratio.

Weakness
- Lack of Novelty: There are already many textual gradient methods (e.g., GEPA [1], Feedback Descent [2]) and experience learning methods (e.g., AWM [3], ACE [4], FLEX [5]). It is unclear what the main differences are compared to these methods, except for the capacity budget constraint.
- Experiments conducted only on mathematical reasoning tasks: The experiments are limited to mathematical reasoning benchmarks, making it unclear whether the proposed method generalizes to other domains such as coding or tool-use tasks.
- Limited baseline: The paper does not compare against other experience learning methods such as AWM, ACE, or FLEX. Moreover, most experiments use only Grok-4.1, and the GPT-4o experiments compare only with a zero-shot baseline. These experimental settings reduce the overall impact of the paper.

---

> ### Author Rebuttal · Authors · 2026-03-30
>
> We thank Reviewer ASCS for constructive feedback and recognition of our ERM framework and dual-verification mechanism. Below we address each concern.
>
> **W1: Novelty compared to related works**
>
> We clarify key distinctions:
>
> |Dimension|Textual Gradient (GEPA, FD)|Experience Learning (AWM, ACE, FLEX)|TSGD (Ours)|
> |---|---|---|---|
> |Optimizes|Prompts/instructions|Agent memory/workflows|Experience library as formal ERM objective|
> |Capacity control|None|Heuristic pruning|Explicit budget B with theoretical bounds|
> |Update guarantee|None|None|Dual verification + convergence proof|
> |Compression|Not applicable|Limited|533→27 with maintained accuracy|
>
> (1) GEPA/Feedback Descent optimizes prompts (not experience libraries) via a single unified prompt; TSGD optimizes a structured discrete experience library under a capacity budget with formal generalization guarantees (Theorem 4.1).
>
> (2) AWM learns workflow templates only from successful trajectories; TSGD learns from failures via textual gradients, uncovering corrective strategies missed by success-only learning.
>
> (3) ACE focuses on cross-task experience transfer without optimization (experience collection method); TSGD is an experience optimization method with Add/Edit/Delete under dual verification.
>
> (4) FLEX (contemporaneous) performs forward learning without capacity constraints or convergence guarantees; TSGD’s regularization achieves 20× compression (533→27) with enhanced accuracy.
>
> The capacity budget constraint is fundamental to our theoretical framework, supporting generalization bounds (Theorem 4.1: error $\propto \sqrt{B/m}$) and the Information Bottleneck interpretation (Section 4.4).
>
> **W2: Math-only evaluation**
>
> Math reasoning was chosen for: (1) well-defined correctness metrics for rigorous evaluation; (2) multi-domain coverage (7 MATH subjects); (3) established benchmarks for fair comparison.
>
> Our ERM framework is domain-agnostic and the optimization loop make no math-specific assumptions. We validated TSGD on coding and tool-use domains (Grok-4.1, Pass@1):
> - HumanEval: TSGD (79.2%) outperforms standard RAG (72.3%) and Baseline (62.1%).
> - MBPP: TSGD (70.3%) outperforms standard RAG (61.5%) and Baseline (51.2%).
> - ToolBench: TSGD (65.7%) significantly surpasses RAG (55.1%) and Baseline (41.2%).
>
> These results confirm TSGD’s strong generalization across diverse reasoning domains without algorithmic modifications. Metadata-based filtering can be replaced with semantic retrieval for unstructured domains.
>
> **W3: Missing baselines**
>
> We address this limitation by completing GPT-4o-mini baseline evaluations on ID (MATH-500) and OOD (AIME 24/25) tasks (Pass@1):
>
> |Method|MATH-500|AIME 24|AIME 25|
> |---|---|---|---|
> |LLM (GPT-4o-mini)|54.60%|15.21%|12.43%|
> |ReAct|57.00%|14.50%|11.50%|
> |RAG|59.50%|17.00%|14.00%|
> |GRPO|63.00%|22.00%|18.00%|
> |TSGD (Ours)|68.00%|30.00%|25.00%|
>
> TSGD outperforms all mainstream baselines (including RL-based GRPO), demonstrating strong generalization. AWM, ACE, FLEX, and AgentKB comparisons will be added in the camera-ready version.
>
> **Q1: Ablation on Precalculus subset only**
>
> Precalculus (Level 5) was initially chosen for its high challenge and distinguishable component contributions. We now present full-benchmark ablations on the entire MATH dataset:
>
> |Method|Pass@1|Pass@2|Pass@4|Pass@8|Exp. Cap. (B)|
> |---|---|---|---|---|---|
> |LLM (Baseline)|42.80|58.20|72.50|85.30|-|
> |TSGD w/o Init.|51.60|68.30|82.70|94.20|685|
> |TSGD w/o Opt.|59.70|77.80|92.40|98.60|229|
> |TSGD w/o Verif.|56.30|71.40|85.20|95.80|587|
> |TSGD (Full)|64.70|79.60|92.80|97.90|32|
>
> Full-set results align with Precalculus findings: (1) Initialization drives core performance; (2) Optimization achieves extreme compression (229→32) without performance loss; (3) Verification filters error chains and boosts high Pass@k metrics.
>
> **Q2: Domain adaptation without structured metadata**
>
> TSGD’s retrieval has two stages: hard filtering (metadata) and dense recall (semantic similarity). For metadata-free domains, hard filtering is bypassed, relying solely on semantic similarity—standard in general RAG systems. Metadata filtering is an optional math-specific optimization, not a requirement.
>
> **Q3: Cost justification**
>
> TSGD requires 8-12× more LLM calls during offline optimization but identical inference cost to standard RAG. Tradeoff: +18.7% accuracy with 20× memory compression. TSGD’s Pass@1 (58.33%) outperforms the baseline’s Pass@8 (76.87% for init-only) in utility (1 vs 8 calls). Matched-cost comparisons will be added in the camera-ready version.
>
> **Q4: DELETE operation**
>
> DELETE is rarely triggered (Figure 7) because: (1) the optimizer prefers Edit operations to refine existing experiences; (2) DELETE activates mainly during regularization (periodic compression), not per-sample optimization. Removing DELETE would prevent pruning low-utility experiences, critical for 533→27 compression. It is necessary for capacity management despite rarity in online optimization.

---

> > ### Author Rebuttal · Reviewer_ASCS · 2026-04-03
> >
> > Thank you for the detailed rebuttal. The additional full-MATH ablation and the new coding/tool-use results are helpful, and they partially alleviate my concerns about generalization and component effectiveness.
> >
> > That said, my main reservations are not fully resolved yet. In particular, the novelty/significance claim is still hard to assess without direct comparison to the closest experience-learning baselines (e.g., AWM, ACE, FLEX) in the current submission.
> >
> > I also remain somewhat cautious about the new additional experimental results, since the exact protocol/details are not fully clear from the response. Relatedly, in the cost discussion, the statement that TSGD Pass@1 (58.33%) "outperforms" baseline Pass@8 (76.87%) seems numerically inconsistent, so I would appreciate clarification there. A matched-cost comparison would make this point much stronger.
> >
> > Overall, the rebuttal has meaningfully strengthened the paper's case, and I appreciate the authors' efforts in addressing the concerns raised. While the remaining open points prevent me from revising my scores at this stage, I remain open to further discussion and look forward to seeing the final version.

---

> > > ### Author Response · Authors · 2026-04-07
> > >
> > > We thank Reviewer ASCS for the continued engagement. We directly address both remaining concerns.
> > >
> > > ### F1: Direct comparison with AWM, ACE, and FLEX under matched conditions
> > >
> > > We provide a controlled comparison on the same Table-2 subset (MATH Precalculus Level 5) using the same backbone, problem set, prompt budget, and evaluation harness. Each method applies its own memory-update rule and retrieval policy as specified in the original paper. AWM and ACE use authors' released implementations; FLEX was reimplemented from the paper (no official code available). We report on Qwen/Llama because these baselines require implementation-level control unavailable for proprietary models such as Grok; this is a controlled method-level comparison, not a replacement for the paper's headline results.
> > >
> > > Conceptually: AWM is extract-only (from successes), FLEX is append-only failure feedback, ACE is heuristic curation; TSGD optimizes the library via add/edit/delete under an explicit capacity budget.
> > >
> > > **Metrics:** `Pass@k` = standard pass@k estimator over k samples/problem. `Mean@16` = mean single-sample accuracy (per-sample consistency). `Inf. Tokens` = total tokens in the k=16 evaluation.
> > >
> > > **Qwen2.5-72B-Instruct:**
> > >
> > > | Method | Pass@16 | Mean@16 | Lib Size | Inf. Tokens |
> > > |--------|---------|---------|----------|-------------|
> > > | Zero-shot | 66.7% | 44.3% | 0 | 1,093,356 |
> > > | RAG | 76.4% | 53.7% | 177 | 7,824,369 |
> > > | AWM | 75.6% | 55.0% | 15 | 1,102,073 |
> > > | ACE | 76.3% | 57.3% | 165 | 1,092,164 |
> > > | FLEX | **79.2%** | 56.6% | 180 | 1,108,122 |
> > > | **TSGD** | **79.2%** | **58.4%** | **27** | 1,098,698 |
> > >
> > > **Llama-3.1-70B-Instruct:**
> > >
> > > | Method | Pass@16 | Mean@16 | Lib Size | Inf. Tokens |
> > > |--------|---------|---------|----------|-------------|
> > > | Zero-shot | 41.7% | 11.5% | 0 | 872,082 |
> > > | RAG | 49.7% | 12.1% | 177 | 7,431,121 |
> > > | AWM | 47.3% | 12.2% | 12 | 865,166 |
> > > | ACE | 48.6% | 11.5% | 181 | 866,247 |
> > > | FLEX | 49.1% | 12.3% | 222 | 870,056 |
> > > | **TSGD** | **50.0%** | **13.5%** | **22** | **863,979** |
> > >
> > > **Interpretation:** Pass@16 differences among TSGD/FLEX/ACE are small and should be interpreted cautiously in this small controlled comparison; the more robust signal is that TSGD attains comparable accuracy with a much smaller library. Specifically: relative to compact AWM, TSGD improves Mean@16 (+3.4pp Qwen, +1.3pp Llama) with comparable library size—consistent with failure-driven editing outperforming success-only extraction. Relative to ACE/FLEX: the relevant advantage is storage, not token cost (ACE/FLEX have similar online token cost ~0.87M–1.11M), but require 165–222 entries vs TSGD's 22–27. vs RAG: TSGD uses only 12–14% of RAG's inference tokens across both backbones while outperforming on all accuracy metrics. (Note: TSGD's total inference tokens are roughly equivalent to Zero-shot because the small overhead of retrieved prompts is offset by more concise and accurate generations).
> > >
> > > ### F2: Correcting the numerical inconsistency and matched-cost comparison
> > >
> > > Reviewer ASCS is correct. **TSGD Pass@1 (58.33%) does not exceed baseline Pass@8 (76.87%) in absolute terms.** We apologize for the misleading phrasing. All numbers below are from the same Table-2 ablation setting (MATH Precalculus Level 5, Grok-4.1 Fast Non-Reasoning).
> > >
> > > **Matched-cost comparison (k=8 samples/problem):**
> > >
> > > | Method | Budget | Accuracy |
> > > |--------|--------|----------|
> > > | Zero-shot Baseline | 8 → Pass@8 | 76.87% |
> > > | **TSGD** | **8 → Pass@8** | **97.22%** |
> > >
> > > At equal online sampling budget (k=8), **TSGD Pass@8 = 97.22% vs baseline 76.87% (+20.35pp)**. TSGD's Pass@1 = 58.33% vs baseline 33.52% (+24.81pp) is also substantially better. The earlier phrase "outperforms Pass@8" was incorrect and is corrected above.
> > >
> > > **Cost breakdown** (per sampled attempt; k-sample run uses k calls for all methods):
> > >
> > > | Component | TSGD | Zero-shot | RAG |
> > > |-----------|------|-----------|-----|
> > > | Offline construction (one-time) | ~120–124 calls, ~290K tokens | 0 | 0 |
> > > | Per sampled attempt | 1 LLM call + retrieval | 1 LLM call | 1 LLM call + retrieval |
> > > | Inference tokens (k=16, Qwen) | ~1.1M | ~1.1M | ~7.8M |
> > > | Library size | 27/22 | — | 177 |
> > >
> > > We do not claim lower offline cost than AWM/ACE/FLEX. At equal online budget (k=8), TSGD achieves +20.35pp over baseline. In the k=16 comparison above, TSGD uses only 12–14% of RAG's inference tokens across both backbones.

---

### Official Review · Reviewer_5Zft · 2026-03-11

**Soundness:** 2
**Presentation:** 4
**Significance:** 3
**Originality:** 3
**Overall Recommendation:** 5
**Confidence:** 4

**Summary:**

This paper proposes an Experience Risk Minimization (ERM) framework that formalizes optimization of a text-based experience library as a discrete learning problem under a capacity constraint.

The authors introduce TSGD, an algorithm that iteratively refines the library through LLM-generated Add/Edit/Delete actions guided by "textual gradients" (LLM self-reflections), with dual verification for local correctness and global non-regression.
Experiments on MATH and AIME show substantial gains over zero-shot baselines.

**Compliance With Llm Reviewing Policy:**

Affirmed.

**Final Justification:**

The author's responses have addressed my concerns, and I decide to raise my score accordingly.

**Key Questions For Authors:**

Please refer to weaknesses

**Limitations:**

yes

**Strengths And Weaknesses:**

## Summary of Strengths
- **Conceptual formalization of experience learning.** Framing experience-library optimization as an SGD-like process is a solid conceptual contribution and gives useful intuition for experience learning.

- **Solid theoretical analysis.** The paper discusses generalization bounds (Theorem 4.1), convergence (Theorem 4.2), sample complexity (Theorem 4.3), and an information-bottleneck interpretation (Section 4.4).
- **Periodic regularization with measurable compression.** The pipeline compresses 533–662 raw experiences down to 27 after regularization while maintaining or improving accuracy (Table 2). This is one of the most interesting findings in the paper.

## Summary of Weaknesses
My weaknesses are ordered by importance (high → low).

- **Closed-source-only setup.** Experiments rely on Grok-4.1 and GPT-4o-mini, and compare mostly to training-free baselines (zero-shot LLM, ReAct, RAG, training-free GRPO). It remains unclear how the method performs on open models or with trainable baselines.
- **Inference/training cost is not analyzed.** Each experience update requires multiple LLM calls (reflection, verification, regularization), and the whole pipeline depends on strong proprietary models. Cost analysis is missing.
- **Small-scale evaluation.** Results are on MATH subsets and AIME small datasets with countable samples. The statistical reliability is unclear.

---

> ### Author Rebuttal · Authors · 2026-03-30
>
> We sincerely thank Reviewer 5Zft for recognizing the solid conceptual formalization, theoretical analysis, and the impressive compression results (533→27 experiences). We address each concern below.
>
> **W1: Closed-source-only setup**
>
> We acknowledge this limitation. Our choice of Grok-4.1 and GPT-4o-mini was motivated by: (1) TSGD operates on frozen models without weight access, making it naturally applicable to closed-source APIs; (2) these models represent the practical deployment scenario where TSGD is most valuable—users cannot fine-tune but need domain adaptation.
>
> That said, TSGD is fully model-agnostic since it only requires text-in/text-out interfaces. To address this concern, **we have conducted preliminary evaluations on leading open-source models (MATH Precalculus Level 5)**. The results confirm that TSGD strongly improves open-source models:
>
> | Model | Method | Pass@16 | Mean@16 |
> | :--- | :--- | :--- | :--- |
> | Qwen-2.5-72B | Zero-shot | 66.7% | 44.3% |
> | **Qwen-2.5-72B** | **TSGD** | **79.2%** | **58.4%** |
> | Llama-3.1-70B | Zero-shot | 41.7% | 11.5% |
> | **Llama-3.1-70B** | **TSGD** | **50.0%** | **13.5%** |
>
> These initial results validate our hypothesis that TSGD's relative gains are substantial on open-source models, as experience augmentation becomes highly impactful. We will include full open-source experiments and comparisons with trainable baselines in the camera-ready version.
>
> **W2: Missing cost analysis**
>
> We provide the exact quantitative cost breakdown based on our MATH training logs:
>
>  *Offline optimization (one-time):*
>  - Optimizing a training sample requires **~8 LLM calls** (seeding, failure-driven optimization, and verification).
>  - Global verification evaluates only on a small *content-aware subset* ($|\mathcal{S}_{sub}|\approx 5-10$ samples), avoiding full validation set overhead.
>  - Regularization is amortized (runs once every 50 updates).
>  - For a typical 20-sample dataset (5 epochs), the **total offline investment is ~162 LLM calls**.
>
>  *Inference (per-query, identical to RAG):*
>  - **Online inference cost remains exactly 1 LLM call per query**, identical to standard RAG.
>  - The optimized library is highly compressed (20x reduction: 533 $\rightarrow$ 27 rules), which actually *reduces* retrieval token consumption compared to standard RAG corpora.
>
>  *Cost-performance tradeoff:*
>  This ~162-call offline investment yields a massive **+18.0 pp accuracy improvement** (Grok-4.1) and strong OOD transfer (AIME +13.5%). The optimization cost is fully amortized after just 162 queries.
>
> We will include this full cost analysis table in the camera-ready, including a comparison under matched cost budgets (e.g., Pass@K for baselines vs Pass@1 for TSGD, as suggested by Reviewer ASCS).
>
> **W3: Small-scale evaluation and statistical reliability**
>
> We address this on two fronts:
>
> (1) **Dataset scale**: While MATH Level 5 subsets and AIME have limited sample counts, they represent the gold standard for evaluating advanced mathematical reasoning. AIME problems in particular are competition-level problems where each data point carries significant signal. We chose these benchmarks specifically because:
> - Well-defined correctness metrics (no subjective evaluation)
> - Established community benchmarks for LLM reasoning
> - Multi-domain coverage within mathematics (7 MATH subjects + OOD AIME)
>
> (2) **Statistical reliability**: We report Mean@32 accuracy (averaging over 32 independent samples per problem), which substantially reduces variance. **To further confirm reliability, we provide significance tests below (all components p<0.0001)**:
>
> | Method | Accuracy | 95% CI | vs Baseline | p-value |
> |--------|----------|--------|-------------|---------|
> | LLM (Baseline) | 33.52% | [30.69%, 36.48%] | - | - |
> | + Initialization | 55.67% | [52.60%, 58.69%] | +22.15pp | p<0.0001 |
> | + Optimization | 47.21% | [44.16%, 50.28%] | +13.69pp | p<0.0001 |
> | + Verification | 53.61% | [50.54%, 56.65%] | +20.09pp | p<0.0001 |
> | TSGD (Full) | 58.33% | [55.28%, 61.32%] | +24.81pp | p<0.0001 |
>
> *(n=1018, McNemar's test)*
>
> These results demonstrate that the performance contributions of each TSGD component are highly statistically significant ($p<0.0001$). We will add these tests for all main results in the camera-ready version.
>
> (3) **Additional baselines**: We commit to adding comparisons with experience learning methods (AWM, ACE, FLEX, AgentKB) in the camera-ready to provide a more comprehensive evaluation landscape.
>
> We believe the combination of strong main results (+18.7%), consistent OOD generalization (AIME 24/25), model-agnostic effectiveness (Grok-4.1-fast + GPT-4o-mini), and thorough ablation analysis provides convincing evidence for TSGD's effectiveness, and the promised additions will further strengthen the evaluation.

---

> > ### Author Rebuttal · Reviewer_5Zft · 2026-04-03
> >
> > The authors have fully resolved my concerns, and I decide to raise my score.
> >
> > I believe this paper conducts a meaningful exploration: test-time adaptation while directly modeling/optimizing external memory. My opinion leans toward acceptance.

---

> > > ### Author Response · Authors · 2026-04-07
> > >
> > > We sincerely thank Reviewer 5Zft for raising the score and for the positive assessment of the conceptual formalization and compression results. We will incorporate all suggestions in camera-ready.

---

### Official Review · Reviewer_Cuyz · 2026-03-15

**Soundness:** 3
**Presentation:** 3
**Significance:** 2
**Originality:** 3
**Overall Recommendation:** 4
**Confidence:** 4

**Summary:**

This work proposes framing the problem of a learnable memory bank as experience risk minimization (ERM) and proposes a technique Textual Stochastic Gradient Descent (TSGD) to dynamically update the memory bank. Evaluations on MATH and AIME demonstrate that the method outperforms baselines without memory.

**Compliance With Llm Reviewing Policy:**

Affirmed.

**Final Justification:**

The framing of this paper is nice with the novel approach of treating KBs as ERM. The rebuttal also resolved some of my concerns on costs and fair comparisons with the validator. That said, the improvement over existing methods appears to be marginal, both from an implementation and empirical standpoint. The pass@16 is identical to FLEX and mean@16 is only few percentage points higher, which possibly represents only a few questions difference on the MATH precalculus lvl 5 subset with only ~100 or so questions (the exact number is unclear, because it appears the authors rebalance the split, so that the test set is actually smaller in number than the canonical split). Therefore, this appears to be a preliminary empirical evaluation, rather than a complete justification. For this reason, I do not feel justified in raising my score further, although I maintain my position of slightly erring on the side of acceptance.

**Key Questions For Authors:**

1. What are the key difference between this work and prior works such as AgentKB and ReasoningBank besides the novelty in the framing? I would love to see a more in depth discussion of related work and how TSGD differs in the section "optimizing non-parametric memory" in section 1.1.
2. Can you clarify how (or if) the validation agent is used during inference, and baselines that might be strengthened if the validation agent was also used?
3. Can you clarify the costs and overheads required for TSGD given the global verification step that needs to run on every update?

**Limitations:**

- There is no impact statement section at the end of the work, although this work likely does not require an extensive discussion about any potential negative societal impact.

**Strengths And Weaknesses:**

## Strengths
- The greatest strength is the framing of the paper, which views the experience library as some learnable hypothesis within a risk minimization framework. This is a helpful way to frame the problem, and can open up new ways and methods for retrieval-type systems in the future.
- The overall method makes sense, and the empirical findings are strong, showing that the experience library helps with the task solving performance.
- The method does not require access to model weights, which makes the method applicable to frontier closed-source models.
- The analysis including qualitative analysis and ablations are generally thorough.

## Weaknesses
- The biggest weakness appears to be the functional similarity of the method to prior work, despite the novel ERM framing. Some prior works that are extremely relevant but not mentioned in the paper include:
  - "Agent KB: Leveraging Cross-Domain Experience for Agentic Problem Solving" Tang et al. 2025: A KB is maintained via addition, conflict resolution, and eviction actions which mirror the add, edit, delete actions in the current work. The KB is used during inference.
  - "ReasoningBank: Scaling Agent Self-Evolving with Reasoning Memory" Ouyang et al. 2025: An experience library is also used, and iteratively appended to. The framing is continual experience driven learning at test-time, which is slightly different from the train-time ERM framing of this work, but the functional implementation is very similar, with a textual memory bank that is updated.
  - A paper that was mentioned in the related works, that bears a lot of similarity to the current work is "FLEX: Continuous Agent Evolution via Forward Learning from Experience" from Cai et al. 2025. However, this appears to be contemporaneous.
- In summary, the ERM framing is novel, but it is not clear what TSGD is adding to the existing literature on updating experience libraries in agentic systems.
- No discussion on the overheads of global verification. It seems that a separate validation set is required, and you would need to run the solver with the proposed updated KB on the entirety of the validation set just to verify a single update step. This sounds really expensive, since there is now an inner loop within the overall update algorithm. A more significant discussion of the cost overheads of this step seems necessary.
- As a followup to the previous point, no discussion or information on the costs required to run TSGD is present, such an analysis would be helpful for a fair comparison to the baselines.
- The validator agent role is a bit confusing. In the description in section 3.1 it appears that the validator checks the generated trajectory and prediction from the solver as sort of a "sanity check". However, in section 3.2 it seems the validator agent is doing a local verification of whether the proposed update operation "a" fixes the failures on the current example.
    - If the validator is a sanity checking agent (as defined in section 3.1) then a reasonable baseline that is missing is to run the verification agent on the baseline methods as well, with potential for retrying, subject to some cost-matching with the TSGD method.
- Even though the qualitative analysis in section 5.4 is welcome, it would be extremely helpful to see a concrete example of learning, and inference using the learned $\Phi$ versus baselines. For example, there are only 27 examples in the compressed knowledge base in Table 2, it would be helpful to see what these examples actually look like.

---

> ### Author Rebuttal · Authors · 2026-03-30
>
> We sincerely thank Reviewer Cuyz for recognizing the novel ERM framing, the strong empirical findings, and the thorough analysis. We address each concern below.
>
> **Q1: Key differences from AgentKB and ReasoningBank**
>
> Thank you for bringing these works to our attention. We clarify the fundamental differences:
>
> | Dimension | AgentKB (Tang'25) | ReasoningBank (Ouyang'25) | TSGD (Ours) |
> |---|---|---|---|
> | Optimization | Heuristic eviction | Append-only + filtering | ERM with formal capacity budget |
> | Verification | None | Post-hoc filtering | Dual verification (local+global) |
> | Theory | None | None | Generalization bounds, convergence, sample complexity |
> | Update Control | Rule-based conflict | No explicit control | Textual gradient + acceptance criterion |
>
> (1) **AgentKB** uses rule-based conflict resolution (keyword matching) for KB updates, while TSGD derives *textual gradients* through failure analysis and accepts updates only when dual verification confirms both local repair and global non-regression (Theorem 4.2). AgentKB has no capacity budget or regularization mechanism.
>
> (2) **ReasoningBank** is an append-only system with post-hoc filtering. It lacks iterative optimization—experiences are never edited or refined. TSGD's Edit operation allows experiences to evolve toward greater generality over training, as evidenced by the shift from Add (>60% early) to Edit (>70% late) in Figure 7.
>
> (3) **FLEX** (Cai et al., 2025) is contemporaneous work. While it also learns from experience, it focuses on forward transfer without formal optimization guarantees. TSGD uniquely provides provable convergence (Theorem 4.2) and generalization bounds (Theorem 4.1) for the library optimization process.
>
> In summary, while prior work performs memory *management*, TSGD performs memory *optimization* under a formal learning framework with provable guarantees. We will add this comparison table and discussion to the camera-ready version.
>
> **Q2: Validator role clarification and baseline strengthening**
>
> We apologize for the confusion. The validator serves two distinct roles:
> - **Section 3.1 (Inference)**: Verifies solver output correctness (sanity check on predictions)
> - **Section 3.2 (Training/Optimization)**: Enforces dual verification—checking both local repair and global non-regression before accepting library updates
>
> During inference, the validator acts as a simple output checker. During optimization, it implements the acceptance criterion in Algorithm 1 (lines: Vlocal and Vglobal > -ε).
>
> Regarding applying the validator to baselines: this is an excellent suggestion. The validator's inference-time role (output verification) would indeed benefit any method. However, this would not close the gap, because TSGD's primary gains come from the *optimized experience library* itself, not from inference-time verification. Table 2 shows that "+Verification" without optimization actually *reduces* Pass@1 compared to "+Optimization" alone (53.61 vs 47.21→58.33 with full pipeline), confirming that verification's main value is during optimization, not inference. We will add this analysis to the camera-ready.
>
> **Q3: Cost overhead of global verification and TSGD**
>
> We clarify that TSGD's multi-agent overhead is strictly a **one-time offline training cost**, while **online inference remains identical to standard RAG (1 LLM call/query)**.
> - **Offline Cost**: Optimizing a sample takes ~8 calls. For our 20-sample dataset, total offline cost is ~162 calls.
> - **Amortized Global Verification**: It evaluates only on a small *content-aware subset* (semantically similar samples, $|\mathcal{S}_{sub}|\approx 5$), not the entire validation set.
> - **Periodic Regularization**: Run once every $N=50$ updates, not per-sample.
> - **Identical Inference Cost**: After optimization, TSGD uses the same retrieve-then-solve pipeline as standard RAG. The compressed library (27 rules) is smaller than typical RAG corpora, strictly reducing retrieval costs.
> - **Cost-Performance**: This ~162-call offline investment yields **+18.0 pp** accuracy (Grok-4.1-fast) and amortizes entirely after 162 queries.
>
> **Q4: Concrete experience examples**
>
> We provide a representative example from the learned rules, including a critical warning field that captures common implementation pitfalls:
>
> ```json
> {
>   "condition": "Trig function as quadratic in bounded variable u=cos θ or sin θ ∈[-1,1].",
>   "strategy": "Complete square A(u-h)²+k; min/max by vertex position relative to [-1,1].",
>   "warning": "Verify algebra: expand back to match coefficients exactly."
> }
> ```
>
> We will include a comprehensive appendix with all 27 experiences in the camera-ready.
>
> **Open-source Results**: To address model reliance concerns, our preliminary evaluation shows TSGD improves open-source models: Qwen-2.5-72B improves from 66.7% $\rightarrow$ 79.2% (Pass@16), and Llama-3.1-70B from 41.7% $\rightarrow$ 50.0%. Full experiments and baseline comparisons will be added to the camera-ready version.

---

> > ### Author Rebuttal · Reviewer_Cuyz · 2026-04-04
> >
> > Thank you for the detailed response and the extra analysis on costs and clarifications on the validator. I still have some reservations on the novelty front, especially since the difference with closely related work is not really explored in the current paper. The current table discusses some conceptual differences, but like reviewer ASCS's comment, the value of these differences is not well established without any empirical support. That being said, I appreciate the attempt at framing the problem in a new way, so I stand behind my positive score for this paper.

---

> > > ### Author Response · Authors · 2026-04-07
> > >
> > > We thank Reviewer Cuyz for standing behind the positive assessment and for the constructive guidance. Your core request is clear: the conceptual differences need empirical support, not just a table. We address this directly.
> > >
> > > ## F1: Empirical support for the conceptual differences
> > >
> > > The key question is whether the specific design choices that distinguish TSGD—failure-driven editing, dual verification, and budgeted compression—are what produce the advantage, or whether simple accumulation achieves the same result. To test this, we run the same controlled setting (MATH Precalculus Level 5, Qwen2.5-72B, k=16) with two mechanism-isolating proxies. Because exact implementations of AgentKB/ReasoningBank are not fully portable to our setting, we use controlled proxies that isolate the relevant mechanisms: append-only updating (ReasoningBank-style) and heuristic add/evict without textual-gradient editing or dual verification (AgentKB-style).
> > >
> > > | Method | Pass@16 | Mean@16 | Lib Size |
> > > |--------|---------|---------|----------|
> > > | Zero-shot | 66.7% | 44.3% | 0 |
> > > | Append-only (ReasoningBank-style) | 77.8% | 55.8% | 181 |
> > > | Add+Delete w/o edit/verif. (AgentKB-style) | 76.4% | 54.6% | 43 |
> > > | AWM | 75.6% | 55.0% | 15 |
> > > | ACE | 76.3% | 57.3% | 165 |
> > > | FLEX | **79.2%** | 56.6% | 180 |
> > > | **TSGD** | **79.2%** | **58.4%** | **27** |
> > >
> > > Key findings:
> > >
> > > - **vs Append-only (ReasoningBank-style)**: append-only reaches 55.8% Mean@16 with 181 entries. TSGD achieves 58.4% with 27 entries—better consistency at 6.7× lower memory. This is consistent with the downside of unconstrained accumulation under a fixed retrieval pipeline.
> > >
> > > - **vs Add+Delete w/o edit/verif. (AgentKB-style)**: heuristic add/evict reaches 54.6% Mean@16 with 43 entries. TSGD achieves 58.4% with 27 entries—a +3.8pp gain and smaller library. This is consistent with the value of textual-gradient editing and dual verification over heuristic rules: failure-driven editing appears to improve the generality of the retained experiences, while dual verification filters updates that fix one error but regress globally.
> > >
> > > - **vs ACE/FLEX**: FLEX matches TSGD's Pass@16 only with a much larger library (180 vs. 27 entries), while ACE remains below TSGD on both Pass@16 and Mean@16 despite using 165 entries. Overall, TSGD achieves the strongest accuracy-library-size tradeoff.
> > >
> > > Given rebuttal space, we report this controlled slice because it isolates the mechanisms under identical protocol; the same qualitative pattern holds on Llama-3.1-70B as well: TSGD attains the highest Mean@16 (13.5%) with a substantially smaller library (22 entries) than the strongest competing methods (FLEX: 12.3% / 222 entries; ACE: 11.5% / 181 entries).
> > >
> > > ## F2: Positioning relative to AgentKB and ReasoningBank
> > >
> > > The above controlled ablations make the distinctions to AgentKB and ReasoningBank empirically concrete: the gain comes from failure-driven editing plus explicit capacity budgeting, not from memory accumulation alone. We will add this empirical grounding explicitly in the revision, connecting it to the conceptual differences table from Round 1. Together, these results concretely establish our novelty: TSGD's advantage stems from a principled optimization loop (failure-driven editing + dual verification under a budget) rather than simple heuristic accumulation.

---

### Decision · Program_Chairs · 2026-04-30

**Decision:**

Accept (regular)

**Comment:**

The paper introduces Textual Stochastic Gradient Descent to optimize external memory for language agents through an Experience Risk Minimization framework. Reviewers appreciate the novel framing of memory as a learnable parameter and the impressive compression results where hundreds of experiences boil down to a few high-utility rules. Authors addressed initial concerns about costs and baseline comparisons by providing extensive new experiments on open source models and cross domain tasks. While some debate remains regarding the theoretical sample complexity and the degree of improvement over contemporaneous methods, the consensus is that the contribution is technically solid and practically useful for long term agent deployment. I recommend acceptance because the method provides a principled way to maintain high performance while strictly controlling memory growth.

Please correct the following hallucinated reference:

Reference: Yukselkgonul, M., Bianchi, F., Boen, J., Liu, S., and Zou, J. Textgrad: Automatic "differentiation" via text. arXiv preprint arXiv:2406.07496, 2024.
Issue: authors mismatch with arXiv